# Data-Aware and Scalable Sensitivity Analysis for Decision Tree Ensembles

**Namrita Varshney, Ashutosh Gupta, Arhaan Ahmad, Tanay V. Tayal, & S. Akshay**
Department of Computer Science and Engineering,
Indian Institute of Technology Bombay, Mumbai, India.
`namrita@iitb.ac.in, akg@cse.iitb.ac.in, arhaan.ahmad2003@gmail.com,`
`tanaytayal@cse.iitb.ac.in, akshayss@cse.iitb.ac.in`

## Abstract

Decision tree ensembles are widely used in critical domains, making robustness and sensitivity analysis essential to their trustworthiness. We study the feature sensitivity problem, which asks whether an ensemble is "sensitive" to a specified subset of features - such as protected attributes- whose manipulation can alter model predictions. Existing approaches often yield examples of sensitivity that lie far from the training distribution, limiting their interpretability and practical value. We propose a data-aware sensitivity framework that constrains the sensitive examples to remain close to the dataset, thereby producing realistic and interpretable evidence of model weaknesses. To this end, we develop novel techniques for data-aware search using a combination of mixed-integer linear programming (MILP) and satisfiability modulo theories (SMT) encodings. Our contributions are four-fold. First, we strengthen the NP-hardness result for sensitivity verification, showing it holds even for trees of depth 1. Second, we develop MILP-optimizations that significantly speed up sensitivity verification for single ensembles and, for the first time, can also handle multiclass tree ensembles. Third, we introduce a data-aware framework that generates realistic examples close to the training distribution. Finally, we conduct an extensive experimental evaluation on large tree ensembles, demonstrating scalability to ensembles with up to 800 trees of depth 8, achieving substantial improvements over the state of the art. This framework provides a practical foundation for analyzing the reliability and fairness of tree-based models in high-stakes applications.

## 1 Introduction

Decision tree ensembles are a popular AI model, known for their simplicity, power, and interpretability. They are ubiquitous across multiple industries, ranging from banking (Chang et al., 2018; Madaan et al., 2021) and healthcare (Ghiasi & Zendehboudi, 2021; Kelarev et al., 2012) to water resources engineering (Niazkar et al., 2024) and telecommunication (Shrestha & Shakya, 2022). Given that this class of models forms a cornerstone for automated decision-making in various industries, it is important to be able to trust their answers and provide guarantees on their reliability. Towards this goal, there has been significant research in the past decade on formalizing and verifying various safety properties of tree ensembles.

In this paper, we focus on one such problem: understanding the influence that a particular subset of input features can have on the output of a decision tree classifier. This notion of *sensitivity* of the model to a feature set has been studied in various contexts in previous works. It has been related to individual fairness and causal discrimination (Dwork et al., 2011; Calzavara et al., 2022; Galhotra et al., 2017; Blockeel et al., 2023), which are central to building responsible AI systems. A model is called sensitive to a specified set of features if the output of the model can be changed by keeping every other feature the same and varying only the specified input features. Thus the problem of feature sensitivity verification (or simply sensitivity) is to check whether a given tree ensemble model $\mathcal{E}$ is sensitive to a specified subset of features $F \subseteq \mathcal{F}$, i.e, whether there exist two inputs, called a *(sensitive) counterexample pair* that are identical on $\mathcal{F} \setminus F$, but on which $\mathcal{E}$ gives different outputs. Knowledge of sensitivity to specific subsets of input features is important for understanding

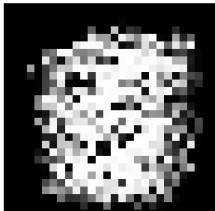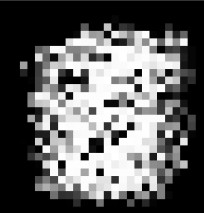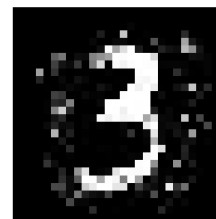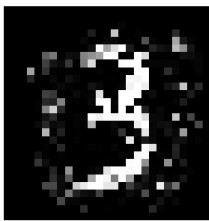

Figure 1: Two counterexample pairs from a tree ensemble trained on MNIST. (Left) A counterexample pair where the left image is classified as 3 and the right as 8; but both are meaningless blobs. (Right) A pair closer to the training distribution. The left image is classified as 3 and the right as 8; where both resemble a 3, but the second is confidently misclassified, providing a more useful witness of sensitivity.

and mitigating attacks that aim to change model outputs by manipulating a small set of protected input features. This analysis can also help uncover unwanted patterns in the trained models that may arise from social biases in the training data.

Recently, Ahmad et al. (2025) showed that sensitivity verification is NP-hard for ensembles with trees of maximum depth at least 3, and gave a tool utilizing a pseudo-Boolean encoding to tackle the problem for binary classifiers. However, their NP-hardness proof (via a 3-SAT reduction) does not extend to ensembles of trees with depth at most 1 or 2, leaving the hardness of the sensitivity problem open for such ensembles. Trees of depth 1, also called decision stumps, have been long studied in the literature Wang et al. (2020); Horváth et al. (2022); Martínez-Muñoz et al. (2007) and are of interest in several applications Chen et al. (2023); Huynh et al. (2018).

A second, major challenge is that sensitive counterexample pairs may a priori lie far from actual data points, providing only weak evidence of a model's sensitivity. To see an illustrative example of this, consider Figure 1, where a tree ensemble trained on MNIST with 786 features yields two counterexample pairs. In the left pair, the decision flips (3 to 8), but neither image likely appears in the training set, so this does not reveal a model weakness. In contrast, the right pair is informative: the first image closely resembles a real training image (3) and is correctly classified, but modifying 20/786 features causes misclassification to 8 while remaining near the data distribution. Do similar cases occur when $|F| = 1$? They do, even in tabular datasets, as illustrated in Section C in the Appendix. This raises the question: can we identify sensitive counterexample pairs that are closer to the real data distribution, enabling meaningful conclusions about model sensitivity? Such counterexamples are valuable for downstream tasks, such as retraining or hardening the model, but in this work we focus solely on their identification, which is already a challenging problem.

We start by showing that the sensitivity problem is NP-hard even for ensembles of decision tree stumps (trees of depth 1) via a novel reduction from the subset-sum problem. Next, to find counterexample pairs closer to the data, we develop two complementary strategies - one using a product of marginal distributions as an objective function and another constraint-solver based approach where we avoid regions of sparsely populated data during the search. The pseudo-Boolean approach of Ahmad et al. (2025) is difficult to extend with such objective functions and hence we revisit a mixed integer linear program (MILP) approach for sensitivity verification. However, a baseline MILP implementation, based on the original encoding of Kantchelian et al. (2016) performs worse than the pseudo-Boolean method, highlighting the challenge of this approach. We introduce novel optimizations to the MILP encoding that result in a significant speed up, making it feasible to analyze large ensembles, while guiding the search toward meaningful counterexample pairs (i.e., close to the data distribution). We also show that our new MILP encoding can be extended to obtain to-the-best-of-our-knowledge the first tool for sensitivity verification over multiclass tree ensemble classifiers. Empirically, we demonstrate the effectiveness of our approach on ensembles trained using XGBoost (Chen & Guestrin, 2016), achieving an order of magnitude improvement in runtime compared to earlier methods, as well as higher quality counterexamples, measured by their proximity to the data distribution. Thus, our main contributions are:

- We show that sensitivity verification is NP-hard even for ensembles of depth-1 trees.

- We significantly advance sensitivity verification by enabling discovery of counterexample pairs closer to the training distribution, through two complementary strategies: one using a product-of-marginals objective, and another a novel constraint-solving based approach to compute clause summaries and prune data-sparse regions in the input space.

- We design a MILP-based encoding with key novel optimizations for sensitivity verification, implemented via a combination of MILP and SMT solvers, and - to the best of our knowledge - are the first to extend sensitivity verification to multiclass decision tree ensembles.

- We implement our approach in a tool ENSENSE and perform extensive experiments on 18 datasets and 36 tree ensembles for binary and multiclass settings. ENSENSE can verify tree ensembles with up to 800 trees of depth 8, significantly outperforming the state of the art.

**Related Work.** A closely related problem is local robustness, which involves finding adversarial perturbations that can cause misclassification. In the context of decision trees, this problem was originally defined in Kantchelian et al. (2016) who showed its NP-hardness and used an MILP encoding to solve it. Since then, a rich line of work has emerged for robustness verification (Devos et al., 2021; Chen et al., 2019a; Ranzato & Zanella, 2020; Törnblom & Nadjm-Tehrani, 2019; Wang et al., 2020), using different techniques, from input-output mappings in Törnblom & Nadjm-Tehrani (2019) to abstract interpretation in Ranzato & Zanella (2020) to dynamic programming Wang et al. (2020) and clique-based approaches in the state-of-the-art tool, VERITAS (Devos et al., 2021). Most recently, Devos et al. (2024) extended the last approach to local robustness verification for multiclass tree-ensembles. While specific ideas from robustness verification are useful for sensitivity verification (and we do build on some of them), the locality of the robustness problem allows a mixture of simplifying optimizations given the knowledge of one input. In contrast, sensitivity verification involves a universal quantification over two inputs, making it a more complex problem.

## 2 PRELIMINARIES

In a classification problem, we are given an input space $\mathcal{X} \subseteq \mathbb{R}^d$ defined over a $d$-dimensional feature space $\mathcal{F}$, and an output space $\mathcal{Y} = \{0, 1, \ldots, C - 1\}$, where $C$ is the number of classes. We intend to learn the unknown mapping $\mathcal{E} : \mathcal{X} \to \mathcal{Y}$. For any $x \in \mathcal{X}$, we will denote the value of feature $f \in \mathcal{F}$ for $x$ as $x_f$. A decision tree is recursively defined as either a leaf node or an internal node. Each leaf node $n$ has a leaf value $n.val$, which is a scalar in $\mathbb{R}$ (for binary classification) or a vector in $\mathbb{R}^C$ (for multiclass classification). Each internal node $n$ consists of references to child nodes, decision trees $n.yes$ and $n.no$, and a guard $n.guard$, which is a linear inequality of the form $X_f < \tau$. Here, $f$ is a feature, $X_f$ denotes the variable for feature $f$, and $\tau$ is a constant. An input $x \in \mathcal{X}$ is evaluated on the tree $T$ by a top-down traversal. For each encountered internal node $n$, the guard of $n$, say $n.guard = X_f < \tau$ is evaluated by substituting $X \leftarrow x$ in the inequality. If the guard inequality evaluates to true, we move to $n.yes$; otherwise, we move to $n.no$. This process continues till we reach a leaf node $n$, and the output of the tree $T(x)$ is given by $n.val$.

To increase the power of a single decision tree, it is common to use ensembling, where multiple decision trees are trained, and the outcomes are aggregated to reach a final decision. Formally, a tree ensemble classifier $\mathcal{E} : \mathcal{X} \to \mathcal{Y}$ consists of a set $\mathcal{T}$ of decision trees. The output of the ensemble is found by aggregating the outputs of each decision tree. There are three notions of outputs from a tree ensemble: (i) $\mathcal{E}_c^{raw}(x)$, which represents a linear aggregation of the outputs of the ensemble for class $c$, typically a sum over the outputs of all trees in the ensemble, (ii) $\mathcal{E}_c^{prob}(x)$: the predicted class probability of class $c$ and (iii) the output label $\mathcal{E}(x) = \arg\max_c \mathcal{E}_c^{prob}(x)$. In the binary classification setting, where $\mathcal{Y} = \{0, 1\}$, each leaf node stores $n.val \in \mathbb{R}$ and $\mathcal{E}_1^{raw}(x) = \sum_{T \in \mathcal{T}} T(x)$, $\mathcal{E}_0^{raw}(x) = -\mathcal{E}_1^{raw}(x)$, $\mathcal{E}_1^{prob}(x) = \text{SIGMOID}(\mathcal{E}_1^{raw}(x))$, $\mathcal{E}_0^{prob}(x) = 1 - \mathcal{E}_1^{prob}(x)$, where SIGMOID $: \mathbb{R} \to \mathbb{R}$ is the sigmoid function defined as $\text{SIGMOID}(x) = 1/(1 + e^{-x})$. For convenience, a glossary of the notation used throughout the paper is provided in Appendix A.

## 3 FEATURE SENSITIVITY AND HARDNESS

In this section, we define the sensitivity verification problem and provide our improved hardness results. Given a decision tree ensemble classifier $\mathcal{E}$, our goal is to find two points $x^{(1)}$ and $x^{(2)}$ in the

input space $\mathcal{X}$, such that they differ only in a specified subset of features $F \subseteq \mathcal{F}$ and have the same values for all the remaining features, while producing different output labels when passed to the classifier, i.e., $\mathcal{E}(x^{(1)}) \neq \mathcal{E}(x^{(2)})$. This problem becomes more significant if, not only are their output labels different, but the predicted class probabilities of the outputs are also far apart, indicating a change from a highly confident positive prediction to a highly confident negative prediction.

**Definition 3.1.** Given a tree ensemble for binary classification $\mathcal{E} : \mathcal{X} \to \{0, 1\}$, a set of features $F \subseteq \mathcal{F}$ and a parameter $g \geq 0$, $\mathcal{E}$ is said to be $(g, F)$-sensitive if we can find two inputs $x^{(1)}, x^{(2)}$ (called a counterexample pair) such that $(x^{(1)})_{\mathcal{F} \setminus F} = (x^{(2)})_{\mathcal{F} \setminus F}$ and $\mathcal{E}_1^{prob}(x^{(1)}) \geq 0.5 + g$ and $\mathcal{E}_1^{prob}(x^{(2)}) \leq 0.5 - g$. The sensitivity verification problem asks if a given tree ensemble classifier $\mathcal{E}$ is $(g, F)$-sensitive.

In what follows, we often fix $g$ and refer to $F$-sensitivity; when $F$ is clear from context, we simply write sensitivity. Ahmad et al. (2025) showed that sensitivity is NP-Hard for tree ensembles with maximum depth $\geq 3$ for $|F| = 1$, $|F| = $ constant and $F = \mathcal{F}$. However, as noted by the authors, their reduction (from 3-SAT) does not work for depth 1 and 2. Our first result is to close this gap by showing NP-hardness at depth 1 using a novel reduction from the subset sum problem, a well-known NP-complete problem Garey & Johnson (1979).

**Theorem 3.2.** *Sensitivity verification with $|F| = 1$ is NP-Hard for tree ensembles with depth $\geq 1$.*

*Proof.* Consider an instance of the integer subset sum problem, i.e., we are given a set of $n$ integers $\mathcal{U}$ and an integer $k$, and we want to find a subset $U \subseteq \mathcal{U}$, such that $\Sigma_{l \in U} = k$. We call the $i^{th}$ integer $u_i$ where $i$ varies from 0 to $n - 1$. For every index $i$, we create a Boolean feature $f_i$. Then we create a decision tree of depth 1, which splits on $f_i$, giving output 0 if $f_i$ is false and $u_i$ otherwise. We create another Boolean feature $f'$ and a decision tree of depth 1, which splits on $f'$, giving output $-k - 0.5$ if $f'$ is false and $-k + 0.5$ otherwise. We call the ensemble of all these trees to be $\mathcal{E} : \{0, 1\}^{n+1} \to \{0, 1\}$ with the trees being $T_i$ where $i$ varies from 0 to $n - 1$ and $T'$. We claim that $\mathcal{E}$ is $\{f'\}$-sensitive iff there exists $U \subseteq \mathcal{U}$ such that $\sum_{l \in U} l = k$. With this claim, the theorem immediately follows. A formal proof of the claim is given in Appendix B. $\square$

We remark that when $|\mathcal{F}/F|$ is bounded, we can solve the depth 1 problem in polynomial time (see Appendix B). This completes the complexity-theoretic picture for the sensitivity problem.

## 4 DATA-AWARE SENSITIVITY VERIFICATION

Sensitivity, as defined in Definition 3.1, requires finding a counterexample pair showing sensitivity, but does not specify how close the inputs $x^{(1)}$ and $x^{(2)}$ in the pair must be to real-world data. Indeed, this is not surprising, as the definition is itself independent of data (including training data), and only depends on the trained model. But, as a consequence, counterexample pairs may include inputs that are arbitrarily far from the true data distribution, as illustrated in Figure 1 in the Introduction. Additional examples are in Appendix C. Our goal, therefore, is to find more meaningful counterexample pairs, towards which we extend Definition 3.1 with a utility function.

**Definition 4.1.** Given a (binary) tree ensemble classifier $\mathcal{E} : \mathcal{X} \longrightarrow \{0, 1\}$, a set of sensitive features $F \subseteq \mathcal{F}$, a gap parameter $g \geq 0$ and a data distribution/utility function $u : \mathcal{X} \times \mathcal{X} \to [0, 1]$, we say that $\mathcal{E}$ is $(g, F, u)$-sensitive, if there exist two inputs $x^{(1)}, x^{(2)} \in \mathcal{X}$ such that $x^{(1)}_{\mathcal{F} \setminus F} = x^{(2)}_{\mathcal{F} \setminus F}$, $\sigma(\mathcal{E}(x^{(1)})) \geq 0.5 + g$, $\sigma(\mathcal{E}(x^{(2)})) \leq 0.5 - g$ and $u(x^{(1)}, x^{(2)})$ is maximal among all such pairs.

We could also add a threshold parameter $\epsilon \in [0, 1]$ and require $u(x^{(1)}, x^{(2)}) \geq \epsilon$. Typically, $u$ serves as a proxy for how similar $x^{(1)}, x^{(2)}$ are to the training distribution. Given a (possibly training) dataset $\mathcal{D}$, we want the utility function $u : \mathcal{X} \times \mathcal{X} \to [0, 1]$ to represent the likelihood of the input pair being drawn from or close to $\mathcal{D}$. In this work, we investigate two approaches to achieve this.

**Utility Function.** For simplicity, we first assume that all input features are independent. This allows us to calculate the likelihood of each feature independently and then multiply the results. For a given feature $f$, consider the guards in the ensemble that involve $f$. Suppose a feature $f$ appears in $K_f$ guards, with sorted thresholds $\tau_{f1} < \cdots < \tau_{fK_f}$. We assume that the $X_f$ takes value within range $[\tau_{f1}, \tau_{fK_f})$, which we can ensure by introducing guards $X_f < -\infty$ and $X_f < \infty$. This partitions

the space of feature $f$ into $K_f - 1$ intervals. We estimate the marginal probability of $f$ lying in each interval $[\tau_{f(k-1)}, \tau_{fk})$ by calculating the count of points in $\mathcal{D}$ for which $f$ lies in $[\tau_{f(k-1)}, \tau_{fk})$ and dividing this by the total count of points in $\mathcal{D}$. That is, for any feature $f$ and real value $v$, we have,

$$\pi_f(v) = \sum_{k=2}^{K_f} \left( \mathbf{1}_{(\tau_{f(k-1)} \leq v < \tau_{fk})} \cdot \frac{|\{x \in \mathcal{D} \mid \tau_{f(k-1)} \leq x_f < \tau_{fk})\}|}{|\mathcal{D}|} \right)$$

And for any input $x = (x_1, x_2, x_3, \ldots, x_d)$, assuming independence of features, we define $\pi(x) = \pi_{f_1}(x_1) \cdot \pi_{f_2}(x_2) \ldots \pi_{f_n}(x_d)$, where $\pi : \mathcal{X} \to [0, 1]$ is the product distribution estimated from $\mathcal{D}$. With this the utility function just becomes $u(x^{(1)}, x^{(2)}) = \pi(x^{(1)}) \cdot \pi(x^{(2)})$.

Intuitively, under the independence assumption, it measures how likely inputs are to be drawn from the distribution $\mathcal{D}$, guiding our approach to look for more meaningful counterexamples. In the next section, we show how this can be encoded effectively, and our experiments indicate that in many benchmarks it does give better counterexample pairs (i.e., closer to data). However, its effectiveness diminishes in datasets with high feature dependencies, which motivates an orthogonal approach.

**Restricting search space using clause summaries.** Our second approach attempts to compute *summaries* of the input space that describe *cavities* - regions where no data points exist. Our goal is to ensure that these cavities are excluded from our sensitivity search. For simplicity, we focus on cavities represented as a bounded boxes in the input space. Given a value of $w$ (a width parameter), we create the following template for points in $\mathcal{D}$ that fall in a cavity:

$$\bigwedge_{i \in [1,w]} (X_{f_i} \geq lb_i \wedge X_{f_i} < ub_i) \qquad (1)$$

where each $lb_i$ and $ub_i$ take values from one of the guards associated with feature $f_i$ appearing in the tree ensemble. In Figure 2, we illustrate a green cavity in 2-D space, where all the data points are projected in dimensions $f_i$ and $f_j$. Therefore, we avoid finding sensitivity pairs in

Figure 2: There are no training set data points within the green box.

$(X_{f_i} \geq lb_i \wedge X_{f_i} < ub_i) \wedge (X_{f_j} \geq lb_j \wedge X_{f_j} < ub_j)$. The difficulty is to find such cavities in the data set. For this, we observe that given a box which is a conjunction of interval regions, its negation is a clause, in a form that can be processed by a constraint solver. Hence, our main idea is to find such cavities in the input data-set using a state-of-the-art Satisfiability Modulo Theories (SMT) solver De Moura & Bjørner (2008), as we detail in Section 5.3.

## 5 AN IMPROVED MILP ENCODING

We build on the MILP encoding for decision trees introduced by Kantchelian et al. (2016). The encoding, when used directly for sensitivity verification, is less efficient than the pseudo-Boolean approach as shown in Ahmad et al. (2025). However, we develop *novel* optimizations to the encoding for sensitivity analysis, which allow the MILP encoding to outperform the pseudo-Boolean encoding by a large margin, achieving an order-of-magnitude runtime reduction compared to SENSPB.

**Base Encoding.** The encoding represents the decision tree ensemble as a set of linear inequalities. It uses a set of binary variables $p_{fk}$ to denote the predicates that appear on the internal node's guard is true, and a set of continuous variables $0 \leq l_n \leq 1$ to denote which leaf node is visited in each tree. The output of the tree ensemble is then computed as a linear combination of the leaf values, weighted by the predicate variables. For each input feature $f$, we ensure consistency across the predicate variables corresponding to the feature, since if $\tau_1 < \tau_2$, then $X_f < \tau_1$ implies $X_f < \tau_2$. Let the corresponding predicate variables be $p_{f1}, p_{f2}, \ldots, p_{fK_f}$. We require that $p_{f1} = 1 \implies p_{f2} = 1$, $p_{f2} = 1 \implies p_{f3} = 1 \ldots p_{f(K_f-1)} = 1 \implies p_{fK_f} = 1$. We encode this in MILP as Eq. (2).

$$p_{f1} \leq p_{f2} \leq \cdots \leq p_{fK_f} \qquad (2) \qquad\qquad l_1 + l_2 + \cdots + l_N = 1 \qquad (3)$$

Let $l_1, l_2, \ldots, l_N$ be the leaf variables corresponding to a tree. We require two leaf consistency conditions. First, we require that exactly one of these leaf variables is set to 1 and every other leaf variable is set to 0, which can be enforced as in Eq. (3). Second, we require that if a leaf variable is set to 1, then every predicate variable in the path to the leaf node needs to be set such that the path

is followed. For each internal node $n$ of $t$th tree, consider the set $TSet$ of leaf nodes in the subtree rooted at $n.yes$ and set $FSet$ of the leaf nodes of subtree rooted at $n.no$. Let $X_f < \tau_{fk}$ be the guard of node $n$ (recall that $X_f$ refers to variables while $x_f$ are concrete input values). If $n$ is a root node, then we add the constraint given in Eq. (4), and for any non-root node, the constraint in Eq. (5).

$$1 - \sum_{n \in FSet} l_n = p_{fk} = \sum_{n \in TSet} l_n \quad (4) \qquad 1 - \sum_{n \in FSet} l_n \geq p_{fk} \geq \sum_{n \in TSet} l_n \quad (5)$$

The constraints in Eq. (2) to Eq.(5) are from Kantchelian et al. (2016). To model the sensitivity problem, we create two instances of all the variables to encode the runs of the two differentiating inputs given to the tree ensemble. We will add superscripts to indicate the copies. We need to ensure that the two inputs differ only in the specified set of features $F \subseteq \mathcal{F}$. For this, denoting by $\mathcal{V}_F$ the set of all predicate variables such that their guards contain some $f \in F$, we add the constraint in Eq. (6). Finally, for binary trees, $\mathcal{E}^{prob}(x) > 0.5 + g \iff \mathcal{E}^{raw}(x) > \text{SIGMOID}^{-1}(0.5 + g)$. Let us define $\delta = \text{SIGMOID}^{-1}(0.5 + g)$. Because of the symmetry of SIGMOID about $y = 0.5$, we have that $\text{SIGMOID}^{-1}(0.5 - g) = -\delta$. Thus, we introduce the constraint described in Eq. (Gap-bin). Recall that for a leaf node $n$, $n.val$ denotes its value. Let $\mathcal{A}$ be the set of all leaves in all trees.

$$\bigwedge_{p_{fk}^{(1)} \notin \mathcal{V}_{\mathcal{F}}} p_{fk}^{(1)} = p_{fk}^{(2)} \quad (6) \qquad \sum_{n \in \mathcal{A}} l_n^{(1)} n.val \geq \delta \wedge \sum_{n \in \mathcal{A}} l_n^{(2)} n.val \leq -\delta \quad \text{(Gap-bin)}$$

Any feasible solution to these constraints corresponds to two inputs $x^{(1)}$ and $x^{(2)}$, which differ only in the feature set $F$ and produce outputs such that $\mathcal{E}(x^{(1)}) \geq 0.5 + g$ and $\mathcal{E}(x^{(2)}) \leq 0.5 - g$.

**Optimizations to the Encoding.** We now describe the novel optimizations that we develop and prove their corrections. Subsequently, we will also explain how we incorporate data-awareness.

## 5.1 Constraints on Unaffected and Affected Leaves

For each leaf, we call the set of all the guards in the path from the root to the leaf as the ancestry of that leaf. A leaf is called *unaffected* if, for each guard in the ancestry of the leaf, the guard predicate does not contain a feature from the set of varying features. Let $\mathcal{U}$ denote the set of indices of all unaffected leaves. For each such leaf, we add a constraint on the two variables $l_n^{(1)}, l_n^{(2)}$ as defined in Eq. (UnAff). Intuitively, if a leaf is reached in a run of the first input, then it will also be reached in the second. Adding this constraint explicitly helps the solver reach a feasible solution faster, especially in cases where the sensitive feature only affects a small subset of the leaves. This is particularly important for features that are present in guards that are farther away from root nodes. In practice, a large fraction of leaves belong to $\mathcal{U}$.

Next, given that a significant fraction of the leaves are unaffected in practical scenarios, we also add constraints to ensure that Leaf variables that are not in $\mathcal{U}$ (i.e., they correspond to "affected" leaves) are capable of influencing the output. This is done by subtracting the two constraints in Eq. (Gap-bin) and using Eq. (UnAff) to remove any terms corresponding to unaffected leaves, leading to the constraints in Eq. (Aff-bin). This constraint has significantly fewer terms than Eq. (Gap-bin) while capturing its essence, leading to significantly faster running times.

$$\bigwedge_{l_n \in \mathcal{U}} l_n^{(1)} = l_n^{(2)} \qquad \text{(UnAff)} \qquad \sum_{l_n \notin \mathcal{U}} \left( l_n^{(1)} n.val - l_n^{(2)} n.val \right) \geq 2 \times \delta \quad \text{(Aff-bin)}$$

Crucially, as the following theorem shows (with Proof in Appendix D), adding these optimizations does not change the set of feasible solutions.

**Theorem 5.1.** *Let $\mathcal{C}$ denote the MILP equation obtained as a conjunction of Equations (2), (3), (4), (5),(6) and (Gap-bin). The set of feasible solutions of $\mathcal{C}$ and the MILP obtained by conjuncting $\mathcal{C}$ with the constraints induced by Eq. (UnAff) and Eq. (Aff-bin) are equal.*

## 5.2 Objective Function

Modern MILP solvers are built to optimize over an objective function and are highly engineered with multiple heuristics to traverse the search space while being mindful of the objective function. For our setting, we just need to find a single feasible solution to the set of constraints $\mathcal{C}$ as defined in Theorem 5.1. Among these constraints, Eq. (Gap-bin) captures the essence of the two outputs being different and reduces the set of feasible solutions by a significant amount. We can utilize the objective function to amplify the importance of this constraint for the solver by adding a constraint in Eq. Obj-bin), capturing the difference between the two equations in Eq. (Gap-bin).

$$\text{MAX} \sum_{n \in \mathcal{A}} \left( l_n^{(1)} n.val - l_n^{(2)} n.val \right) \tag{Obj-bin}$$

This addition leads to significant improvement in running times, as the objective function can guide the MILP solver in choosing the better edge when faced with multiple candidates during the simplex-solving process instead of arbitrarily choosing between each of the available constraints. In Appendix H, we present the full encoding of an illustrative example.

## 5.3 Modification in MILP for Data-Aware Search

Finally, we modify the MILP encoding to solve data-aware sensitivity as defined in Def. 4.1.

**Utility Function.** Given the utility function as defined in Section 4, we replace the objective function in Eq. (Obj-bin) by the following objective, which maximizes the value of the utility function:

$$\text{MAX} \sum_{f \in \mathcal{F}} \sum_{k=2}^{K_f} (\log(\pi_f(\tau_{f(k-1)})) - \log(\pi_f(\tau_{fk})))(p_{fk}^{(1)} + p_{fk}^{(2)}). \tag{7}$$

Importantly, we convert the product of marginals to log values, as MILP solvers only handle additive constraints on the objective function. The following lemma establishes the correctness of the data-aware objective function given in equation 7.

**Lemma 5.2.** *The objective function in 7 maximizes $u(x^{(1)}, x^{(2)})$.*

*Proof.* Let $p_{fk}^{(1)}$ be true for smallest $k$. Due to the consistency constraints, all $p_{f(k+1)}^{(1)}, ..., p_{fK_f}^{(1)}$ are true. Let $p_{fk'}^{(2)}$ be true for smallest $k'$. Therefore, the sum will reduce to $\sum_{f \in \mathcal{F}} \log(\pi_f(\tau_{f(k-1)})) + \log(\pi_f(\tau_{f(k'-1)})) - 2\log(\pi_f(\tau_{fK_f}))$. We may ignore the last term as it is a constant and we may replace $\tau_{f(k-1)}$ by $x_f^{(1)}$ because $x_f^{(1)} \in [\tau_{f(k-1)}, \tau_{fk})$ and $\tau_{f(k'-1)}$ by $x_f^{(2)}$ because $x_f^{(2)} \in [\tau_{f(k'-1)}, \tau_{fk'})$. Therefore, the total sum will be $\sum_{f \in \mathcal{F}} \log(\pi_f(x_f^{(1)})) + \log(\pi_f(x_f^{(2)}))$, Since the objective function is maximizing the sum, it is maximizing our utility function $u(x^{(1)}, x^{(2)}) = \Pi_{f \in \mathcal{F}} \pi_f(x_f^{(1)}) \cdot \pi_f(x_f^{(2)})$. $\square$

**Computing clause summaries.** Next we define constraints that can identify cavities in data and their negations, i.e., clauses that guide the sensitivity search. In Eq. (1), we need to learn the features and their bounds. Let $r_{if}$ be a Boolean variable indicating $f_i$ is $f$, and $s_{ik}$ indicating $lb_i = \tau_{fk}$, and $t_{ik}$ indicating $ub_i = \tau_{fk}$, where $\tau_{fk}$ is the $k$-th guard for feature $f$. For each $x \in \mathcal{D}$, we add:

$$\bigvee_{i \in [1,w]} \left( \bigwedge_{f \in \mathcal{F}, \, k,k' \in [1,K_f]} (r_{if} \wedge s_{ik} \wedge t_{ik'} \rightarrow ((x_f < \tau_{fk} \vee x_f \geq \tau_{fk'}))) \right)$$

To avoid redundancies, we enforce the ordering of features: $i < j \rightarrow f_i < f_j$. While solving the constraints to find a clause that satisfies all samples, we also add an objective function to guide towards learning tight clauses as: $\text{MIN} \left( \sum_{i \in [1,w]} (\sum_{k=1}^{|K|} k \cdot s_{ik} - \sum_{k'=1}^{|K|} k' \cdot t_{ik'}) \right)$, where $K = max_f(K_f)$. This ensures that we select the smallest guard $k$ for the lower bound of some feature and the largest guard $k'$ for the upper bound of the feature. We iteratively compute one clause at a time and add constraints to exclude solutions corresponding to previously computed clauses.

| Binary classifiers | | | | |
| --- | --- | --- | --- | --- |
| SNO | ModelName | #Trees | Dep. | #Feat. |
| 1 | breast_cancer_robust | 4 | 4 | 10 |
| 2 | breast_cancer_unrobust | 4 | 5 | 10 |
| 3 | diabetes_robust | 20 | 4 | 8 |
| 4 | diabetes_unrobust | 20 | 5 | 8 |
| 5 | ijcnn_robust | 60 | 8 | 22 |
| 6 | ijcnn_unrobust | 60 | 8 | 22 |
| 7 | adult | {200,300,500} | {4,5} | 15 |
| 8 | churn | {200,300,500} | {4,5} | 21 |
| 9 | pimadiabetes | {200,300,500} | {4,5} | 9 |
| 10 | german_credit | {500,800} | {4,5} | 20 |

| Multi classifiers | | | | |
| --- | --- | --- | --- | --- |
| SNO | ModelName | #Trees | Dep. | #Classes |
| 1 | covtype_robust | 100 | 6 | 10 |
| 2 | covtype_unrobust | 100 | 6 | 10 |
| 3 | fashion_robust | 100 | 6 | 10 |
| 4 | fashion_unrobust | 100 | 6 | 10 |
| 5 | ori_mnist_robust | 100 | 6 | 10 |
| 6 | ori_mnist_unrobust | 100 | 6 | 10 |
| 7 | Iris | 100 | 1 | 3 |
| 8 | Red-Wine | 100 | 4 | 5 |

Table 1: Benchmark details. Binary models 1–6 are taken from Chen et al. (2019b); 7–10 are trained on UCI datasets Dua & Graff (2019) using all combination of #Trees $\in \{200, 300, 500\}$ and Depth $\in \{5, 6\}$ (six configurations). Multiclass models 1–6 are from Chen et al. (2019b); models 7–8 are trained on UCI datasets Dua & Graff (2019). Dep. refers to the average depth in the ensemble across all trees of the model.

Initially, we learn clauses of size one and progressively increase the size up to a user-defined limit. The computed clauses provide a summary of $\mathcal{D}$, capturing how the data is distributed across the input space and add the learned clauses to the constraints. Any solution to the query is required to satisfy these clauses, thereby making it more likely to align with the data.

# 6 EXTENSION TO MULTI-CLASS TREE ENSEMBLES

We extend our formalism and encoding to the multi-class setting. Let $\mathcal{Y} = \{0, 1, \ldots, C-1\}$ denote the set of $C$ classes in a multiclass tree ensemble. The set of trees $\mathcal{T}$ is partitioned into $C$ equal partitions, one for each class denoted by $\mathcal{T}_0, \mathcal{T}_1, \ldots, \mathcal{T}_{C-1}$. The trees in a partition $\mathcal{T}_c$ are one-vs-rest classifiers for the class $c$; that is, they consider class $c$ as the positive class, and everything else combined as the negative class and train like a binary classifier. Formally, $\mathcal{E}_c^{raw}(x) = \sum_{T \in \mathcal{T}_c} T(x)$, and $\mathcal{E}_c^{prob}(x) = \text{SOFTMAX}_c\left(\mathcal{E}_0^{raw}(x), \ldots, \mathcal{E}_{C-1}^{raw}(x)\right)$, where $\text{SOFTMAX}_c : \mathbb{R}^C \to \mathbb{R}$ is the softmax function defined as $\text{SOFTMAX}_c(x_0, x_1, ..., x_{C-1}) = e^{x_c}/\Sigma_{k=0}^{C-1} e^{x_k}$. The output is the class with the highest probability, i.e., $\mathcal{E}(x) = Argmax_{c \in \mathcal{Y}} \mathcal{E}_c(x)$. Thus, given a tree ensemble for multiclass classification $\mathcal{E} : \mathcal{X} \to \{0, 1, \ldots, C-1\}$, $c^{(1)}, c^{(2)} \in \{0, 1, \ldots C-1\}$, we find $x^{(1)}, x^{(2)} \in \mathcal{X}$ such that $\mathcal{E}(x^{(1)}) = c^{(1)} \neq \mathcal{E}(x^{(2)}) = c^{(2)}$. We also extend the parameterized version of Def. 3.1 by requiring that the difference between probability of most and second-most probable class is large:

**Definition 6.1.** Given tree ensemble $\mathcal{E} : \mathcal{X} \to \mathcal{Y}$, $F \subseteq \mathcal{F}$, $g \geq 0$, two classes $c^{(1)}, c^{(2)} \in \mathcal{Y}$, $\mathcal{E}$ is $(g, F, c^{(1)}, c^{(2)})$−sensitive if there exist $x^{(1)}, x^{(2)}$ such that $(x^{(1)})_{\mathcal{F} \setminus F} = (x^{(2)})_{\mathcal{F} \setminus F}$, $\forall c \neq c^{(1)}, \mathcal{E}_{c^{(1)}}^{prob}(x^{(1)}) \geq g \times \mathcal{E}_c^{prob}(x^{(1)})$, and $\forall c \neq c^{(2)}, \mathcal{E}_{c^{(2)}}^{prob}(x^{(2)}) \geq g \times \mathcal{E}_c^{prob}(x^{(2)})$.

Now to extend our MILP encoding to the multiclass setting it turns out that we only need to modify Eq. (Gap-bin), Eq. (Aff-bin) and Eq. (Obj-bin) (and its data-aware variant). We present the modifications and prove their correctness in Appendix E.

# 7 EXPERIMENTS

We implement the MILP encoding from Section 5 with all structural optimizations and then add our two data-aware objectives in a tool called ENSENSE. We trained models with XGBoost v1.7.1; we evaluate sensitivity using a single CPU core per run with a per-instance 3600 seconds timeout. We use Gurobi Gurobi Optimization, LLC (2024) as the MILP solver. We focus on single-feature sensitivity; results on varying multiple (viz., 2, 3, 4, and 5) features are presented in Appendix 7. We address the following research questions: **RQ1:** How does our MILP encoding with optimizations fare against the baseline and state-of-the-art for (i) binary and (ii) multi-class classification?

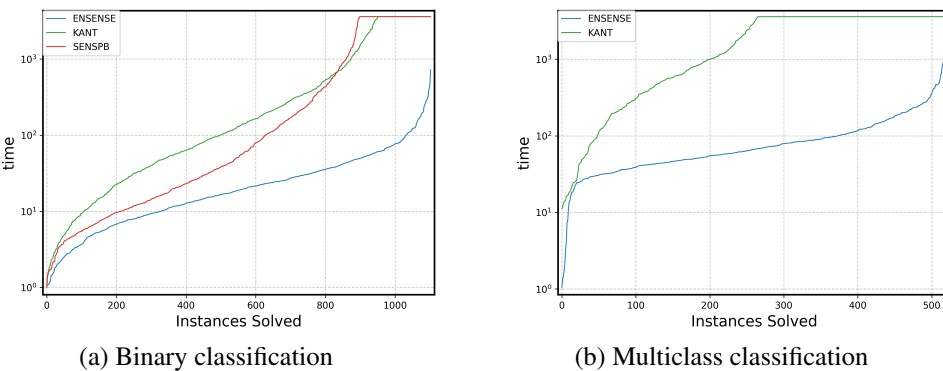

(a) Binary classification  (b) Multiclass classification

Figure 3: Cactus plot comparing runtimes of single feature sensitivity for binary and multiclass.

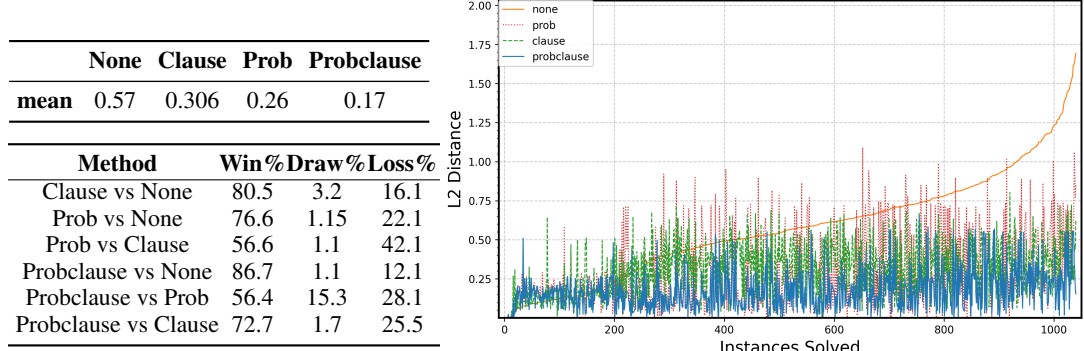

|  | None | Clause | Prob | Probclause |
|---|---|---|---|---|
| **mean** | 0.57 | 0.306 | 0.26 | 0.17 |

| Method | Win% | Draw% | Loss% |
|---|---|---|---|
| Clause vs None | 80.5 | 3.2 | 16.1 |
| Prob vs None | 76.6 | 1.15 | 22.1 |
| Prob vs Clause | 56.6 | 1.1 | 42.1 |
| Probclause vs None | 86.7 | 1.1 | 12.1 |
| Probclause vs Prob | 56.4 | 15.3 | 28.1 |
| Probclause vs Clause | 72.7 | 1.7 | 25.5 |

Figure 4: In top-left, we report the mean $\ell_2$ distance of all instances for each method. In the bottom-left table, we report the win, draw, and loss rates for all pairwise method comparisons. On the right side, a cactus plot of the distance (from the data set to the counterexample found) across binary benchmarks. Instances are sorted by non data-aware baseline (none, orange, solid); for each position, distances of Prob (red, dotted), Clause (green, dashed), and Probclause (blue, solid) are evaluated on same instance (no re-sorting). Lower curves indicate better quality.

**RQ2:** How does data-aware sensitivity perform in giving better quality counterexamples? How do we measure it, what do we compare it against and how do each of our strategies help?

**Binary Classification.** To answer RQ1(i), we compare the performance of ENSENSE against SENSPB the tool using pseudo-Boolean encoding from Ahmad et al. (2025), and KANT, the baseline MILP encoding (adapted from Kantchelian et al. (2016)). We use a wide set of benchmarks mentioned in Table 1 (left), with number of trees ranging from $4-800$, with depth from $4-8$. Considering each single-feature variant as a separate instance and with $gap \in \{0.5, 1, 1.5\}$ this yields a total of 1,290 benchmark instances. Figure 3(a) reports results for the 1,103 instances whose runtime is $\geq 1\,\mathrm{s}$; we omit 187 instances solved in $< 1\,\mathrm{s}$ for fair comparison, which demonstrates the superior performance of ENSENSE. ENSENSE *achieves average speedups of approximately* $8\times$ *over* KANT *and* $5\times$ *over* SENSPB*, with no timeouts whereas* SENSPB *times out for 205 and* KANT *for 153 instances.*

**Multiclass Classification.** For RQ1(ii) we compare ENSENSE against the baseline MILP (still called KANT), as SENSPB does not handle multiclass ensembles. We also repurposed the versatile robustness verification tool VERITAS from Devos et al. (2024) that can handle multiclass ensembles, to solve sensitivity. As it timed out on all except two instances, we detail these results in Appendix F. Table 1(right) lists the multiclass benchmarks. For fashion and ori_mnist, which have 784 features, we restricted to single-feature sensitivity for the 100 most frequently used features in the tree model. For the others, we tested on all features, making a total of 538 benchmark instances. Our results are in Fig3(b), where we again dropped 17 instances which were solved in $< 1$ sec. *The results*

*demonstrate that* ENSENSE *outperforms* KANT *by roughly 15x average speed up, and does not timeout on any of these benchmarks, whereas* KANT *times out on 256.*

**Data-Aware Sensitivity Verification.** For RQ2, to evaluate data-aware sensitivity, we compare (i) the baseline (no data-awareness) with (ii) the utility-based objective that steers solutions toward data-dense regions (called Prob), (iii) the clause-summary strategy that prunes empty-data regions (called Clause), (iv) the combination of both (called Probclause), which ENSENSE uses.

To evaluate the performance of data-awareness sensitivity of each method, we compute the $\ell_2$ distance over insensitive features from the data to the nearest counterexample region identified. Given an input $x$, a *counterexample region* is a connected subset of the input space containing $x'$ such that: (i) the tree ensemble's prediction is *constant* throughout the region (all points fall into the same combination of leaves across all trees), and (ii) every point in the region has a label $y$ with same probability (e.g., is misclassified). Intuitively, tree ensembles partition $\mathbb{R}^d$ into axis-aligned polytopes (one per joint leaf pattern); within each polytope, the model's output does not change. A counterexample region is one of these polytopes (or an intersection thereof with additional constraints) that certifies a whole *set* of violating inputs, not just a single adversarial point. We focus on regions rather than a single point because if the nearest data point lies in the same counterexample region, the correct distance is zero, whereas distance to any particular witness point inside that region can be nonzero and misleading.

Experiments are conducted on the same binary classification benchmarks (from Table 1) with $gap \in \{0.5, 1, 1.5\}$. We used z3De Moura & Bjørner (2008) to synthesize clauses with a maximum of 3 literals, with counterexample-guided refinement, followed by greedy literal-pruning to minimize each clause without reducing coverage. We also discard clauses that enclose an input point, and limit to 1500 synthesized clauses. Our results in Fig 4 show that Utility-based vs. baseline: wins 76.65% of pairs (losses 22.19%, draws 1.15%), with mean distance advantage 0.435 on wins and mean deficit 0.11 on losses. Clause-summary vs. baseline: wins 80.59% (losses 16.13%, draws 3.26%), with mean advantage 0.34 (wins) and mean deficit 0.09 (losses). Combined (Probclause): strongest overall—wins 86.04% of pairs versus baseline, with mean advantage 0.47 on wins and mean deficit 0.06 on losses. *In summary, our results show significant improvement in quality of counterexample pairs (measured by their $\ell_2$ distance from data), with best results obtained by probclause used by* ENSENSE.

We also performed ablation studies on binary and multiclass ensembles to evaluate the contribution of the MILP optimizations, affected and unaffected constraints that we present in Appendix G.2. More experiments and technical details are given in the ArXiv version Varshney et al. (2026) and code is available at https://github.com/formal-trust-AI/ensense.

## 8 CONCLUSION

In this work, we defined a data-aware variant of the sensitivity problem on tree ensembles and developed two approaches to solve this. We developed a new MILP encoding with several improvements, that allows us to improve the quality of the sensitivity witness reported while at the same time providing upto $5\times$ speedup for binary classification over the existing methods and $15\times$ for multiclass classification. One obvious direction for future work is to develop methods for training tree ensembles such that sensitivity can be reduced. This is analogous to the development of various tools for training decision trees that are hardened for local robustness. The other direction for future work is to go from identifying sensitive pairs to quantifying sensitive areas across the input space.

ACKNOWLEDGMENTS

We acknowledge the State Bank of India (SBI) Foundation Hub for Data Science & Analytics , IIT Bombay for supporting the work done in this project. We thank Shri Shakeel Ahmed Agasimani, Deputy General Manager, Analytics Department, C Ramesh Chander, Assistant General Manager (Statistician), Muthukumaran M S, Chief Manager (Data Scientist), Komaragiri Srinivas Jagannath, Manager (Data Scientist), Neha Maheswari, Manager (Data Scientist), State Bank of India, for multiple wide-ranging discussions and feedback. We are grateful to Sunita Sarawagi for her valuable discussions and guidance throughout this work.

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

## APPENDIX

The appendix is organized into 8 sections, in the order in which they occurred in the paper:

- In Section A, we provide a table of notation used throughout the paper for easy reference.
- In Section B, we provide Additional Proofs and theoretical results relevant to Section 3.
- In Section C, we provide data-aware sensitivity examples from Tabular Data as promised in Section 4.
- In Section D, we prove the correctness of our encoding optimizations from Section 4.
- In Section E, we describe the multi-class ensemble encoding into MILP that was left out of Section 5 due to lack of space.
- In Section F, we describe the difficulty with using the VERITAS Devos et al. (2024) framework for comparison and how we can encode our sensitivity problem in that framework and perform some comparisons.
- In Section G, we provide additional experimental setup details as well as additional experiments including:
  - (a) an ablation study for our encoding improvements/optimizations on binary and multiclass tree ensembles
  - (b) a multi-feature sensitivity analysis.
- In section H, we provide an illustrative example of our encoding for a small decision tree ensemble.

# A   NOTATION TABLE

In this section, we provide a table of notation used throughout the paper for easy reference.

| Symbol | Meaning |
|--------|---------|
| $\mathcal{X}$ | input space of the classifiers |
| $\mathcal{Y}$ | output space of the classifiers |
| $\mathcal{F}$ | set of all features |
| $f$ | a feature in $\mathcal{F}$ |
| $x$ | an input in $\mathcal{X}$ |
| $x_f$ | value of feature $f$ for input $x$ |
| $T$ | a decision tree |
| $n$ | a node in a decision tree |
| $n.val$ | leaf value of leaf node $n$ |
| $n.guard = X_f < \tau$ | guard condition of internal node $n$ |
| $n.yes$ | child node of internal node $n$ for true guard evaluation |
| $n.no$ | child node of internal node $n$ for false guard evaluation |
| $T(x)$ | output of tree $T$ on input $x$ |
| $X_f$ | variable for feature $f$ |
| $\mathcal{T}$ | set of decision trees in the ensemble |
| $\mathcal{T}_c$ | set of decision trees in the ensemble for class $c$ |
| $\mathcal{E}$ | a decision tree ensemble |
| $x^{(1)}, x^{(2)}$ | inputs to the ensemble |
| $F$ | set of features to check sensitivity against |
| $g$ | minimum probability threshold for data-aware sensitivity |
| $u(x^{(1)}, x^{(2)})$ | utility function to maximize |
| $\mathcal{D}$ | training data samples |
| $\tau_{fk}$ | threshold in the $k$th guard of feature $f$ |
| $K_f$ | the number of guards for feature $f$ |
| $\pi_f$ | marginal probability function for feature $f$ |
| $w$ | maximum size of the cavity constraints in Eq. 1. |
| $lb_i$ | lower bound on feature $f_i$ in the cavity constraints in Eq. 1 |
| $ub_i$ | upper bound on feature $f_i$ in the cavity constraints in Eq. 1 |
| $p_{fk}$ | Boolean variable for the truth value of $k$th guard of feature $f$ |
| $l_i$ | Variable denoting leaf $i$ is visited. |
| $\delta$ | $Sigmod^{-1}(0.5 - g)$ |
| $\mathcal{V}_F$ | set of all predicate variables for features in $F$ |
| $\mathcal{A}$ | set of all leaf nodes in the ensemble |
| $\mathcal{U}$ | set of all unaffected leaves |
| $r_{fi}$ | Boolean variable to indicate that the feature in $i$th conjunct of cavity is feature $f$ |
| $s_{ik}$ | Boolean variable to indicate that the $i$th conjunct of cavity uses $k$th guard of its feature as lower bound |
| $t_{ik}$ | Boolean variable to indicate that the $i$th conjunct of cavity uses $k$th guard of its feature as upper bound |

# B   ADDITIONAL PROOFS AND THEORETICAL RESULTS

**Theorem B.1.** *The feature sensitivity problem with $|F| = 1$ is NP-Hard for tree ensembles with depth $\geq 1$.*

*Proof.* Consider an instance of the integer subset sum problem, i.e., we are given a set of $n$ integers $\mathcal{U}$ and an integer $k$, and we want to find a subset $U \subseteq \mathcal{U}$, such that $\Sigma_{l \in U} = k$. We call the $i^{th}$ integer $u_i$ where $i$ varies from 0 to $n-1$. For every index $i$, we create a Boolean feature $f_i$. Then we create a decision tree of depth 1 which splits on $f_i$ giving output 0 if $f_i$ is false and $u_i$ otherwise. We create another Boolean feature $f'$ and a decision tree of depth 1 which splits on $f'$ giving output $-k - 0.5$

if $f'$ is false and $-k + 0.5$ otherwise. These trees are depicted in Figure 5. We call the ensemble of all these trees to be $\mathcal{E} : \{0,1\}^{n+1} \to \{0,1\}$ with the trees being $T_i$ where $i$ varies from 0 to $n-1$ and $T'$.

*Claim.* We claim that $\mathcal{E}$ is $\{f'\}$-sensitive iff there exists $U \subseteq \mathcal{U}$ such that $\sum_{l \in U} l = k$.

To see this, consider a function $S : \{0,1\}^{n+1} \to \mathcal{P}(\mathcal{U})$ where $\mathcal{P}(\mathcal{U})$ denotes the power set of $\mathcal{U}$ defined as $S(x) = \{u_i \in \mathcal{U} | x_i = 1 \text{ for some } i \in \{0, 1, \dots, n-1\}\}$. By construction of the trees, $\Sigma_{l \in S(x)} l = \Sigma_{i=0}^{n-1} T_i(x)$ where $x \in \{0,1\}^{n+1}$. If $\mathcal{E}$ is sensitive to $\{f'\}$, then there exist $x^{(1)}$ and $x^{(2)}$ such that $\mathcal{E}^{raw}(x^{(1)}) < 0$ and $\mathcal{E}^{raw}(x^{(2)}) > 0$ and $x^{(1)}_{\perp f'} = x^{(2)}_{\perp f'}$, where $x_{\perp f'}$ refers to input $x$ projected into all features except $f'$. By construction, $x^{(1)}_{f'} = 0$ and $x^{(2)}_{f'} = 1$. Then $\mathcal{E}^{raw}(x^{(1)}) = \Sigma_{i=0}^{n-1} T_i(x^{(1)}) + T'(x^{(1)}) < 0 \implies \Sigma_{l \in S(x^{(1)})} l - k - 0.5 < 0 \implies \Sigma_{l \in S(x^{(1)})} l < k + 0.5$. Similarly, $\mathcal{E}^{raw}(x^{(2)}) = \Sigma_{i=0}^{n-1} T_i(x^{(2)}) + T'(x^{(2)}) > 0 \implies \Sigma_{l \in S(x^{(2)})} l - k + 0.5 > 0 \implies \Sigma_{l \in S(x^{(2)})} l > k - 0.5$. Also, by construction of $S$, $S(x^{(1)}) = S(x^{(2)})$ since $x^{(1)}$ and $x^{(2)}$ only differ on $f'$ and $S$ is independent of that value. Let $S(x^{(1)}) = S(x^{(2)}) = U$. Thus, $k - 0.5 < \Sigma_{l \in U} l < k + 0.5$. As all numbers are integers, $\Sigma_{l \in U} l = k$ and thus there exists $U \subseteq \mathcal{U}$ such that $\Sigma_{l \in U} l = k$.

If there exists $U \subseteq \mathcal{U}$ such that $\Sigma_{l \in U} l = k$ then consider $x^{(1)}$ and $x^{(2)}$ such that $x^{(1)}_i = x^{(2)}_i = 1$ if $u_i \in U$ and $x^{(1)}_j = x^{(2)}_j = 0$ if $u_j \notin U$ where $i, j \in \{0, 1, \dots, n-1\}$. Also, $x^{(1)}_{f'} = 0$ and $x^{(2)}_{f'} = 1$. Note $S(x^{(1)}) = S(x^{(2)}) = U$. Thus, $\mathcal{E}(x^{(1)}) = \Sigma_{l \in S(x^{(1)})} l + T'(x^{(1)}) = k - k - 0.5 = -0.5 < 0$ and $\mathcal{E}(x^{(2)}) = \Sigma_{l \in S(x^{(2)})} l + T'(x^{(2)}) = k - k + 0.5 = 0.5 > 0$. Also $x^{(1)}_{\perp f'} = x^{(2)}_{\perp f'}$. Thus, $\mathcal{E}$ is sensitive to $\{f'\}$ as the above $x^{(1)}$ and $x^{(2)}$ are a required pair of inputs to show sensitivity.

Thus, by proving in both directions, we have shown if we can solve sensitivity for decision tree ensembles of depth-1, then we can solve integer subset sum problem. Thus, sensitivity is at least as hard as integer subset sum and thus it is NP-Hard for depth 1. □

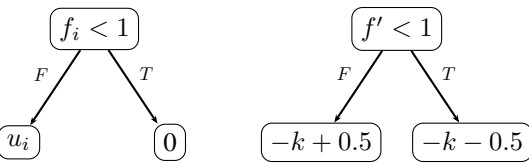

Figure 5: Trees for Proof of Theorem B.1

Finally, we show that when $|\mathcal{F}/F|$ is bounded, we can solve the depth 1 problem in polynomial time.

**Theorem B.2.** *The feature sensitivity problem with bounded $|\mathcal{F}/F|$ is solvable in polynomial time for tree ensembles with depth 1.*

*Proof.* Given a sensitivity problem with decision tree ensemble of trees of depth 1 $\mathcal{E} = \{T_0, T_1, \dots\}$, feature set $\mathcal{F}$, features for checking sensitivity against $F$ and a scalar $k$ such that $|\mathcal{F}/F| \leq k$, we can solve the problem in polynomial time. Let $|\mathcal{E}| = m$ i.e. there are total $m$ decision trees and $|\mathcal{F}| = n$ i.e there a total of $n$ features.

As each tree splits on only 1 feature, we can create a set of trees corresponding to each feature. Thus, we create a function $S(f)$ from the set of feature space to a subset of all trees. Note, it is a partition of all the trees as each tree will split on exactly one feature.

For all $f \in F$, consider the set $S(f)$. We will calculate the minimum and maximum value of sum output of trees in $S(f)$ and their corresponding features values. There are a total of $|S(f)| + 1$ distinct possible values and thus can be found in $O(m)$ time.

We will add the minimums and maximums found above and get a global minimum and maximum value say $m$ and $M$. We need to find whether there exists an assignment to features in $\mathcal{F}/F$ such that the sum of output corresponding to trees of these features lies between $-M$ and $-m$.

For each feature $f \in \mathcal{F}/F$, the possible number of distinct outputs is $|S(f)| + 1$. Therefore, the total possible number of outputs are $\prod_{f \in \mathcal{F}/F}(S(f) + 1)$ which is $O(m^k)$. Thus, by checking for all possible outputs, we can find whether such output exists or not. If it does, then the ensemble is sensitive otherwise it isn't. Thus we have a polynomial time algorithm as $k$ is not a parameter but a bound. □

## C  DATA-AWARE SENSITIVITY EXAMPLES IN TABULAR DOMAINS

This section presents examples that demonstrate the effectiveness of incorporating data-awareness into sensitivity analysis, using models trained on tabular datasets. In each case, we compare sensitive input pairs discovered with and without data-aware methods, showing how the inclusion of data distribution knowledge leads to more realistic counterexamples. Please note that the IJCNN model is trained by Chen et al. (2019b), and the dataset for this model is unfortunately not available with the original feature names. The feature names are simply mentioned as f1 to f22.

*Sensitive Example of IJCCN Chen et al. (2019b)*: In the examples 1 and example 2 below, we analyse the sensitivity with respect to features 3 and 15 in the model trained on the IJCNN dataset. Both methods detect a sensitive pair by varying only these features respectively, resulting in change in the model's prediction. However, the distance from the training data distribution reveals a clear difference: The pair found without data-awareness has a distance of 1.077(example 1) and 1.06 (example 2), indicating it is quite far from any possible realistic data point and may not be very helpful in practice. In contrast, the pair identified with data-awareness has a distance of just 0.327(example 1) and 0.34 (example 2), meaning it is much closer to the data distribution and training data.The insensitive features in the training data points that are far away from the sensitive pair are highlighted with cyan color.

*Sensitive Examples of Adult: ( based on Adult dataset Dua & Graff (2019))*: Here, we present the analysis of the 'adult' dataset in examples 3 and 4, this time examining the sensitivity of the features 'age' and 'sex'. The non-data-aware (baseline) method identifies the sensitive pairs with larger distances of 0.656 and 0.9899 for sensitive features 'age' and 'sex', respectively. The data-aware method, however, identifies pairs that are much closer to the data distances of 0.019 and 0.108. In example 4, the baseline(non-data-aware) method reports an 'age' value of 86, even though the closest datapoint has an age of '40'. The data-aware method, however, identifies a pair with an age value of '46', very close to the nearest datapoint, which has an age of '45'. A similar pattern appears for capital-gain and capital-loss. The baseline method(non-data-aware) selects a pair with values 10,585 (capital-gain) and 3,142 (capital-loss), while the nearest datapoint has values 0 and 2,258. The data-aware method, by comparison, identifies a pair where both features are 0, matching the nearest datapoint exactly. The insensitive features in the training data points that are far away from the sensitive pair are highlighted with cyan color.

*Sensistive Examples of Pimadiabetes: (Dua & Graff (2019))*: Finally, we present one more sensitive pair example, the Pima Diabetes dataset (with no categorical features), in example 5, and examine the sensitivity of feature ('BloodPressure'). Again, the non-data-aware method identifies the sensitive pairs with larger distances of 0.35 , where almost all feature values are far from the data(which we show in cyan color) . However, the data-aware method again identifies the pairs that are much closer to the data distances of 0.03, where only feature "Glucose" is far and the others are close. The examples clearly show that data-aware search finds sensitive pairs closer to the data.

## D  PROOF OF THEOREM 5.1 (CORRECTNESS OF OPTIMIZATIONS)

In this section, we prove Theorem 5.1, which says that our encoding optimiziations, that gave rise to significant improvement in performance, are sound.

*Proof.* We first prove that equation UnAff is already subsumed by equation 2-equation 6. This is established using a proof by contradiction. Assume to the contrary and let $n \in \mathcal{U}$ such that $l_n^{(1)} \neq l_n^{(2)}$. Since the leaf variables are forced to be either 0 or 1 (by equation 4), assume without loss of generality that $l_n^{(1)} = 1$ and $l_n^{(2)} = 0$.

Example 1: IJCNN ROBUST Chen et al. (2019b)

**Sensitive Point Found Without Data-Awareness Analysis For Sensitive Feature 3**

```
Point1: {'f1': 0.0, 'f2': 1.0, 'f3':  0.0, 'f4': 0.0, 'f5': 1.0, 'f6':
    1.0, 'f7': 0.0, 'f8': 0.0, 'f9': 1.0, 'f10': 0.0, 'f11': 0.872834, '
    f12': 1.21062, 'f13': 0.637325, 'f14': 0.356398, 'f15': 0.482769, '
    f16': 0.390789, 'f17': 0.609402, 'f18': 0.558829, 'f19': 0.607626, '
    f20': 0.696077, 'f21': 0.448006, 'f22': 0.619263}
Point2: : {'f1': 0.0, 'f2': 1.0, 'f3':  1.0, 'f4': 0.0, 'f5': 1.0, 'f6':
    1.0, 'f7': 0.0, 'f8': 0.0, 'f9': 1.0, 'f10': 0.0, 'f11': 0.872834,
    'f12': 1.210621, 'f13': 0.637325, 'f14': 0.356398, 'f15': 0.482769,
    'f16': 0.390789, 'f17': 0.609402, 'f18': 0.558829, 'f19': 0.607626,
    'f20': 0.696077, 'f21': 0.448006, 'f22': 0.619263}
Distance from data: 1.07757866169
Nearest Training Datapoint: {'f1': 0.0, 'f2': 1.0, 'f3': 0.0, 'f4': 0.0,
    'f5':  0.0, 'f6':  0.0, 'f7': 0.0, 'f8': 0.0, 'f9':
    0.0, 'f10': 0.0, 'f11':  0.540556, 'f12':  0.250375, 'f13':
    0.467001, 'f14':  0.470297, 'f15':  0.570899, 'f16':  0.527529,
    'f17':  0.517118, 'f18': 0.51931, 'f19':  0.51546, 'f20':  0.55852,
    'f21':  0.53037, 'f22':  0.52894}
```

**Sensitive Point Found With Data-Awareness Analysis For Sensitive Feature 3**

```
Point1: {'f1': 0.0, 'f2': 0.0,  'f3':  0.0, 'f4': 0.0, 'f5': 1.0, 'f6':
    0.0, 'f7': 0.0, 'f8': 0.0, 'f9': 0.0, 'f10': 0.0, 'f11': 0.678628, '
    f12': 0.541327, 'f13': 0.512386, 'f14': 0.516051, 'f15': 0.516459, '
    f16': 0.497491, 'f17': 0.475932, 'f18': 0.495826, 'f19': 0.459603, '
    f20': 0.502477, 'f21': 0.50531, 'f22': 0.507861}
Point2: {'f1': 0.0, 'f2': 0.0,  'f3':  1.0, 'f4': 0.0, 'f5': 1.0, 'f6':
    0.0, 'f7': 0.0, 'f8': 0.0, 'f9': 0.0, 'f10': 0.0, 'f11': 0.678628, '
    f12': 0.541327, 'f13': 0.512386, 'f14': 0.516051, 'f15': 0.516459, '
    f16': 0.497491, 'f17': 0.475932, 'f18': 0.495826, 'f19': 0.459603, '
    f20': 0.502477, 'f21': 0.50531, 'f22': 0.507861}
Distance from data: 0.327039
Nearest Training Datapoint: {'f1': 0.0, 'f2': 0.0, 'f3': 0.0, 'f4': 0.0,
    'f5': 1.0, 'f6': 0.0, 'f7': 0.0, 'f8': 0.0, 'f9': 0.0, 'f10': 0.0,
    'f11':  0.536865, 'f12':
    0.250375, 'f13': 0.51317, 'f14': 0.50743, 'f15': 0.526203, 'f16':
    0.497706, 'f17': 0.484023, 'f18':  0.524588, 'f19':  0.541842,
    'f20':  0.467001, 'f21':  0.470298, 'f22':  0.570898}
```

Example 1: IJCNN ROBUST Chen et al. (2019b)

Example 2: IJCNN ROBUST Chen et al. (2019b)

**Sensitive Point Found Without Data-Awareness Analysis For Sensitive Feature 15**

```
Point1: {'f1': 0.0, 'f2': 1.0, 'f3': 1.0, 'f4': 0.0, 'f5': 1.0, 'f6':
    1.0, 'f7': 0.0, 'f8': 0.0, 'f9': 0.0, 'f10': 1.0, 'f11': 0.822872, '
    f12': 0.276149, 'f13': 0.448491, 'f14': 0.653076, 'f15':
    0.64666, 'f16': 0.615426, 'f17': 0.36238, 'f18': 0.561026, 'f19':
    0.533231, 'f20': 1.199128, 'f21': 0.394238, 'f22': 0.558881}
Point2: {'f1': 0.0, 'f2': 1.0, 'f3': 1.0, 'f4': 0.0, 'f5': 1.0, 'f6':
    1.0, 'f7': 0.0, 'f8': 0.0, 'f9': 0.0, 'f10': 1.0, 'f11': 0.822872, '
    f12': 0.276149, 'f13': 0.448491, 'f14': 0.653076, 'f15':
    0.30856, 'f16': 0.615426, 'f17': 0.36238, 'f18': 0.561026, 'f19':
    0.533231, 'f20': 1.199128, 'f21': 0.394238, 'f22': 0.558881}
Distance from data 1.0647264749
Nearest Training Datapoint:: {'f1': 0.0, 'f2': 1.0, 'f3': 0.0, 'f4':
    0.0, 'f5':  0.0, 'f6':  0.0, 'f7': 0.0, 'f8': 0.0, 'f9': 0.0, 'f10':
    0.0, 'f11':  0.579145, 'f12':  0.18406, 'f13': 0.456363,  'f14':
    0.543959, 'f15':  0.531502, 'f16':  0.449082, 'f17':
    0.483391, 'f18': 0.520606,  'f19':  0.511848, 'f20':  0.527679,
    'f21':  0.474918, 'f22':  0.477095}
```

**Sensitive Point Found With Data-Awareness Analysis For Sensitive Feature 15**

```
Point1 : {'f1': 0.0, 'f2': 1.0, 'f3': 0.0, 'f4': 0.0, 'f5': 0.0, 'f6':
    0.0, 'f7': 0.0, 'f8': 0.0, 'f9': 0.0, 'f10': 0.0, 'f11': 0.797221, '
    f12': 0.341174, 'f13': 0.540222, 'f14': 0.380818,  'f15':
    1.168982, 'f16': 0.457969, 'f17': 0.368609, 'f18': 0.457974, 'f19':
    0.527069, 'f20': 0.399163, 'f21': 0.567169, 'f22': 0.501212}
Point2 : {'f1': 0.0, 'f2': 1.0, 'f3': 0.0, 'f4': 0.0, 'f5': 0.0, 'f6':
    0.0, 'f7': 0.0, 'f8': 0.0, 'f9': 0.0, 'f10': 0.0, 'f11': 0.797221, '
    f12': 0.341174, 'f13': 0.540222, 'f14': 0.380818,  'f15':
    0.412158, 'f16': 0.457969, 'f17': 0.368609, 'f18': 0.457974, 'f19':
    0.527069, 'f20': 0.399163, 'f21': 0.567169, 'f22': 0.501212}
 Distance from data  0.3449
 Nearest Training Datapoint:: {'f1': 0.0, 'f2': 1.0, 'f3': 0.0, 'f4':
    0.0, 'f5': 0.0, 'f6': 0.0, 'f7': 0.0, 'f8': 0.0, 'f9': 0.0, 'f10':
    0.0, 'f11':  0.579295, 'f12':  0.16411, 'f13':  0.44351, 'f14':
    0.45535, 'f15':  0.542591, 'f16':  0.501968, 'f17':  0.500353,
    'f18':  0.5093, 'f19':  0.495474, 'f20':
    0.515701, 'f21': 0.511422, 'f22': 0.517442}
```

Example 2: IJCNN ROBUST Chen et al. (2019b)

---

**Example 3: Adult Dua & Graff (2019)**

**Sensitive Point Found Without Data-Awareness Analysis For Sensitive Feature 'age'**

```
Point1: {'age':66, 'workclass': 'Never-worked', 'fnlwgt': 574792.14018,
    'education': 'Some-college', 'education-num': 16, 'marital-status':
    'Widowed', 'occupation': 'Transport-moving', 'relationship': '
    Unmarried', 'race': 'White', 'sex': 'Male', 'capital-gain': 10585, '
    capital-loss': 2309, 'hours-per-week': 92, 'native-country': 'Poland
    '},
Point2: {'age':86, 'workclass': 'Never-worked', 'fnlwgt': 574792.14018,
    'education': 'Some-college', 'education-num': 16, 'marital-status':
    'Widowed', 'occupation': 'Transport-moving', 'relationship': '
    Unmarried', 'race': 'White', 'sex': 'Male', 'capital-gain': 10585, '
    capital-loss': 2309, 'hours-per-week': 92, 'native-country': 'Poland
    '}
Distance from data : 0.6569890
Nearest Training Datapoint: {'age': 32, 'workclass': 'Private',
    'fnlwgt': 226975, 'education': 'Some-college', 'education-num':
    10, 'marital-status': 'Never-married', 'occupation': 'Sales',
    'relationship':
    'Own-child', 'race': 'White', 'sex': 'Male', 'capital-gain': 0,
    'capital-loss': 1876, 'hours-per-week': 60, 'native-country':
    'United-States'}
```

**Sensitive Point Found With Data-Awareness Analysis For Sensitive Feature 'age'**

```
Point1: {'age': 46, 'workclass': 'Self-emp-inc', 'fnlwgt':
    180532.54372, 'education': 'Doctorate', 'education-num': 13.500002,
    'marital-status': 'Married-civ-spouse', 'occupation': 'Exec-
    managerial', 'relationship': 'Husband', 'race': 'Black', 'sex': '
    Female', 'capital-gain': 0, 'capital-loss': 0, 'hours-per-week': 40,
    'native-country': 'Puerto-Rico'},
Point2: {'age': 33, 'workclass': 'Self-emp-inc', 'fnlwgt':
    180532.54372, 'education': 'Doctorate', 'education-num': 13.500002,
    'marital-status': 'Married-civ-spouse', 'occupation': 'Exec-
    managerial', 'relationship': 'Husband', 'race': 'Black', 'sex': '
    Female', 'capital-gain': 0, 'capital-loss': 0, 'hours-per-week': 40,
    'native-country': 'Puerto-Rico'}
Distance from data 0.0191765741
Nearest Training DataPoint: {'age': 44, 'workclass': 'Private',
    'fnlwgt': 211759, 'education': 'Bachelors', 'education-num': 13, '
    marital-status': 'Married-civ-spouse', 'occupation': 'Exec-
    managerial', 'relationship': 'Husband', 'race': 'Other', 'sex':
    'Male', 'capital-gain': 0, 'capital-loss': 0, 'hours-per-week': 40,
    'native-country': 'Puerto-Rico'}
```

Example 3: Adult Dua & Graff (2019)

---

**Example 4:** Adult Dua & Graff (2019)

**Sensitive Point Found Without Data-Awareness Analysis For Sensitive Feature 'sex'**

```
Point1: {'age': 86, 'workclass': 'Without-pay', 'fnlwgt': 520260.32927,
    'education': 'HS-grad', 'education-num': 16.0, 'marital-status': '
    Never-married', 'occupation': 'Transport-moving', 'relationship': '
    Unmarried', 'race': 'Amer-Indian-Eskimo', 'sex':
    'Female', 'capital-gain': 10585, 'capital-loss': 3142, 'hours-per-
    week': 99, 'native-country': 'Laos'},
Point2: {'age': 86, 'workclass': 'Without-pay', 'fnlwgt': 520260.32927,
    'education': 'HS-grad', 'education-num': 16.0, 'marital-status': '
    Never-married', 'occupation': 'Transport-moving', 'relationship': '
    Unmarried', 'race': 'Amer-Indian-Eskimo', 'sex':
    'Male', 'capital-gain': 10585, 'capital-loss': 3142, 'hours-per-week
    ': 99, 'native-country': 'Laos'}
 Distance from data: 0.98998791099
Nearest Training Datapoint: { 'age':  40, 'workclass':
    'Private','fnlwgt':  287983, 'education':  'Bachelors',
    'education-num':
    13, 'marital-status': 'Never-married', 'occupation':
    'Tech-support', 'relationship':  'Not-in-family', 'race':
    'Asian-Pac-Islander', 'sex': 'Female', 'capital-gain':  0,
    'capital-loss':  2258, 'hours-per-week':  48, 'native-country':
    'Philippines',}
```

**Sensitive Point Found With Data-Awareness Analysis For Sensitive Feature 'sex'**

```
 Point1:{'age': 46, 'workclass': 'Self-emp-inc', 'fnlwgt': 284508.95444,
     'education': 'Doctorate', 'education-num': 13, 'marital-status': '
    Married-civ-spouse', 'occupation': 'Craft-repair', 'relationship': '
    Husband', 'race': 'Black', 'sex':
    'Male', 'capital-gain': 0, 'capital-loss': 0, 'hours-per-week': 50,
    'native-country': 'France'},
 Point2: {'age': 46, 'workclass': 'Self-emp-inc', 'fnlwgt':
    284508.95444, 'education': 'Doctorate', 'education-num': 13, '
    marital-status': 'Married-civ-spouse', 'occupation': 'Craft-repair',
     'relationship': 'Husband', 'race': 'Black', 'sex':
    'Female', 'capital-gain': 0, 'capital-loss': 0, 'hours-per-week':
    50, 'native-country': 'France'}
Distance from data: 0.1081739837784343
Nearest Training Datapoint: {'age': 45, 'workclass':
    'Private', 'fnlwgt':  238567, 'education':  'Bachelors', 'education-
    num': 13, 'marital-status': 'Married-civ-spouse', 'occupation':
    'Exec-managerial', 'relationship': 'Husband', 'race':  'White', 'sex
    ': 'Male', 'capital-gain': 0, 'capital-loss': 0, 'hours-per-week':
    40, 'native-country':  'England'}
```

Example 4: Adult Dua & Graff (2019)

---

Example 5: Pimadiabetes Dua & Graff (2019)

**Sensitive Point Found Without Data-Awareness Analysis For Sensitive Feature 2**

```
Point1: {'Pregnancies': 17, 'Glucose': 188, 'BloodPressure':
    122, 'SkinThickness': 33, 'Insulin': 846, 'BMI': 67.1, '
    DiabetesPedigreeFunction': 2.42, 'Age': 81},
Point2: {'Pregnancies': 17.0, 'Glucose': 188, 'BloodPressure':
    76, 'SkinThickness': 33, 'Insulin': 846, 'BMI': 67.1, '
    DiabetesPedigreeFunction': 2.42, 'Age': 81}
Distance from data: 0.3534358888
Nearest Training Datapoint: {'Pregnancies':  10, 'Glucose':
    148, 'BloodPressure': 84, 'SkinThickness':  48, 'Insulin':  237,
    'BMI': 37.6, 'DiabetesPedigreeFunction':  1.001, 'Age':  51}
```

**Sensitive Point Found With Data-Awareness Analysis For Sensitive Feature 2**

```
Point 1:{'Pregnancies': 2e-06, 'Glucose': 139, 'BloodPressure':70, '
    SkinThickness': 0, 'Insulin': 0, 'BMI': 32.75, '
    DiabetesPedigreeFunction': 0.3595, 'Age': 21}
Point2: {'Pregnancies': 2e-06, 'Glucose': 139, 'BloodPressure':79, '
    SkinThickness': 0, 'Insulin': 0, 'BMI': 32.75, '
    DiabetesPedigreeFunction': 0.3595, 'Age': 21}
Distance from data: 0.03051399
Nearest Training Datapoint: {'Pregnancies': 0, 'Glucose':
    132, 'BloodPressure': 78, 'SkinThickness': 0, 'Insulin': 0, 'BMI':
    32.4, 'DiabetesPedigreeFunction': 0.393, 'Age': 21}
```

Example 5: Pimadiabetes Dua & Graff (2019)

By equation 3, there exists leaf $n'$ such that $l_{n'}^{(2)} = 1$ and the leaves $n$ and $n'$ belong to the same tree. Let $n''$ be the last common node in the paths from the root to leaves $n$ and $n'$, respectively. Let $n''$ be labeled with with $X_f < \tau_k$ and $p_{fk}$ be the corresponding predicate. Since $n''$ is in the ancestry of $n'$ and $n \in \mathcal{U}$, we have that $p_{fk}^{(1)} = p_{fk}^{(2)}$ (by equation 6).

Without loss of generality, assume that leaf $n$ is present in the subtree rooted at $n''.no$, while leaf $n'$ is present in the one rooted at $n''.yes$. Since $l_n^{(1)} = 1$, equation 5 implies that $1 - 1 \geq p_{fk}^{(1)} \implies p_{fk}^{(1)} = 0$. At the same time, since $l_{n'}^{(2)} = 1$, equation 5 implies that $p_{fk}^{(2)} \geq 1 \implies p_{fk}^{(2)} = 1 \neq p_{fk}^{(1)}$ leading to a contradiction. Hence, $l_n^{(1)} = l_n^{(2)} \ \forall i \in \mathcal{U}$ is implied by equation 2-equation 6 and hence the set of feasible solutions does not change on the addition of equation UnAff.

As mentioned earlier, we show that equation UnAff and equation Gap-bin together imply equation Aff-bin. Subtracting the two inequalities in equation Gap-bin, we get that when equation Gap-bin holds, then $\sum l_n^{(1)} n.val - l_n^{(2)} n.val \geq 2 \times \delta$.

However, for the leaves belonging to $\mathcal{U}$, the difference terms are $0$ by definition, i.e. $\sum_{n \in \mathcal{U}} l_n^{(1)} n.val - l_n^{(2)} n.val = 0$. Using these two equations we conclude that if equation UnAff holds and equation Gap-bin holds, then we must have $\sum_{n \notin \mathcal{U}} l_n^{(1)} n.val - l_n^{(2)} n.val \geq 2 \times \delta$, which completes the proof. □

## E  MILP ENCODING FOR MULTICLASS SENSITIVITY

In the main paper, we had extended the Sensitivity problem from binary to multiclass classification. Here we provide the details regarding how we extend the MILP encoding to tackle the multiclass setting. We observe that for the $(g, F, c^{(1)}, c^{(2)})-$sensitivity problem for a multiclass ensemble, only equation Gap-bin, equation Aff-bin and equation Obj-bin need to be modified. We now describe these changes. Let $\mathcal{L}_c$ denote the indices of the leaf variables corresponding to the trees of class $c$.

The change in equation Gap-bin follows from Definition 6.1. To encode that $\forall c \neq c^{(1)}, \mathcal{E}_{c^{(1)}}^{prob}(x^{(1)}) \geq g \times \mathcal{E}_c^{prob}(x^{(1)})$, we instead move to the space of $\mathcal{E}^{Raw}$, i.e. the values before applying SOFTMAX. Given the definition of SOFTMAX,

$$\mathcal{E}_{c^{(1)}}^{prob}(x^{(1)}) \geq g \times \mathcal{E}_c^{prob}(x^{(1)})$$

$$\implies \frac{\text{EXP}(\mathcal{E}_{c^{(1)}}^{raw}(x^{(1)}))}{\sum \text{EXP}(\mathcal{E}_{c_i}^{raw}(x^{(1)}))} \geq g \times \frac{\text{EXP}(\mathcal{E}_{c^{raw}(x^{(1)})})}{\sum \text{EXP}(\mathcal{E}_{c_i}^{raw}(x^{(1)}))}$$

$$\implies \mathcal{E}_{c^{(1)}}^{raw}(x^{(1)}) \geq \mathcal{E}_c^{raw}(x^{(1)}) + \ln g$$

We call $\ln g$ as $\eta$, to get the new gap constraints

$$\bigwedge_{c \neq c^{(1)}} \sum_{l_n \in \mathcal{L}_{c^{(1)}}} l_n^{(1)} n.val > \sum_{l_n \in \mathcal{L}_c} l_n^{(1)} n.val + \eta$$

$$\bigwedge_{c \neq c^{(2)}} \sum_{l_n \in \mathcal{L}_{c^{(2)}}} l_n^{(2)} n.val > \sum_{l_n \in \mathcal{L}_c} l_n^{(2)} n.val + \eta \qquad \text{(Gap-multi)}$$

With this new constraint gap constraint, we can arrive at a constraint for the affected leaves, similar to equation Aff-bin, by adding the two constraints in equation Gap-multi and using reasoning analogous to that of the binary classification setting.

$$\sum_{\substack{l_n \in \mathcal{L}_{c^{(1)}} \\ n \notin \mathcal{U}}} \left( l_n^{(1)} n.val - l_n^{(2)} n.val \right) + \sum_{\substack{l_n \in \mathcal{L}_{c^{(2)}} \\ n \notin \mathcal{U}}} \left( l_n^{(2)} n.val - l_n^{(1)} n.val \right) > 2 \times \eta \qquad \text{(Aff-multi)}$$

An objective function can be formulated as before:

$$\text{MAX} \sum_{l_n \in \mathcal{L}_{c^{(1)}}} \left( l_n^{(1)} n.val - l_n^{(2)} n.val \right) + \sum_{l_n \in \mathcal{L}_{c^{(2)}}} \left( l_n^{(2)} n.val - l_n^{(1)} n.val \right) \qquad \text{(Obj-multi)}$$

**Theorem E.1.** *The set of feasible solutions of the MILP defined by equation $2 \wedge$ equation $3 \wedge$ equation $4 \wedge$ equation $5 \wedge$ equation $6 \wedge$ equation $Gap - multi$ and that of the MILP defined by adding equation Aff-multi are equal.*

The only difficult part in the proof is to see how we obtain equation Aff-multi. Let us choose $c = c^{(2)}$ in the first half and $c = c^{(1)}$ in the second half in equation Gap-multi. We obtain

$$\sum_{l_n \in \mathcal{L}_{c^{(1)}}} l_n^{(1)} n.val > \sum_{l_n \in \mathcal{L}_{c^{(2)}}} l_n^{(1)} n.val + \eta$$

$$\sum_{l_n \in \mathcal{L}_{c^{(2)}}} l_n^{(2)} n.val > \sum_{l_n \in \mathcal{L}_{c^{(1)}}} l_n^{(2)} n.val + \eta \qquad (8)$$

Add the two equations

$$\sum_{l_n \in \mathcal{L}_{c^{(1)}}} (l_n^{(1)} - l_n^{(2)}) n.val + \sum_{l_n \in \mathcal{L}_{c^{(2)}}} (l_n^{(2)} - l_n^{(1)}) n.val > 2\eta \qquad (9)$$

Since for the unaffected leafs $l_n^{(1)} - l_n^{(2)}$ is zero. We derive the desired equation.

$$\sum_{\substack{l_n \in \mathcal{L}_{c^{(1)}} \\ n \notin \mathcal{U}}} \left( l_n^{(1)} n.val - l_n^{(2)} n.val \right) + \sum_{\substack{l_n \in \mathcal{L}_{c^{(2)}} \\ n \notin \mathcal{U}}} \left( l_n^{(2)} n.val - l_n^{(1)} n.val \right) > 2 \times \eta \qquad \text{(Aff-multi)}$$

## F    COMPARISON WITH VERITAS

One of the comments that we left in the main paper was the comparison or lack there-of with the tool VERITAS, a versatile tool for robustness verification of decision tree ensembles, for which there is a multi-class variant available. In this section, we explain why we cannot easily compare with that tool and also how it can be modified so that we can compare it. Firstly, VERITAS does not solve the problem directly as it is not designed for sensitivity. So as the first step, we modified the tool enable the multiclass sensitivity analysis. Note that VERITAS is a more generalizable tool and we approach the problem differently. To encode the multiclass feature sensitivity problem in VERITAS, we create two instances of a given tree ensemble and optimize the following objective:

$$\text{MAX}\left(D_0(x^{(1)}, x^{(2)}) - \text{MAX}_{c,c\neq0}(D_c(x^{(1)}, x^{(2)}))\right) \tag{10}$$

where $D$ is defined as follows:

$$D_c(x^{(1)}, x^{(2)}) = \begin{cases} \mathcal{E}^{raw}_{c^{(1)}}(x^{(1)}) + \mathcal{E}^{raw}_{c^{(2)}}(x^{(2)}), & \text{if } c = 0 \\ \mathcal{E}^{raw}_0(x^{(1)}) + \mathcal{E}^{raw}_0(x^{(2)}), & \text{if } c = c^{(1)} \\ \mathcal{E}^{raw}_{c^{(2)}}(x^{(1)}) + \mathcal{E}^{raw}_{c^{(1)}}(x^{(2)}), & \text{if } c = c^{(2)} \\ \mathcal{E}^{raw}_c(x^{(1)}) + \mathcal{E}^{raw}_c(x^{(2)}), & otherwise \end{cases}$$

We define the objective value found by VERITAS being "better than" ENSENSE if the output of VERITAS is greater than $2 \times \eta$.

Here we provide the proof of correctness of this comparison.

From the equation Gap-multi, we can conclude:

$$\mathcal{E}^{raw}_{c^{(1)}}(x^{(1)}) - \text{MAX}_{c,c\neq c^{(1)}}\mathcal{E}^{raw}_c(x^{(1)}) \geq \eta \tag{11}$$

$$\mathcal{E}^{raw}_{c^{(2)}}(x^{(2)}) - \text{MAX}_{c,c\neq c^{(2)}}\mathcal{E}^{raw}_c(x^{(2)}) \geq \eta \tag{12}$$

$$\tag{13}$$

Ideally we would like to maximize the sum of LHS of the above equations. We will prove that our objective is an upper bound for the output described above.

*Claim.* For all $x^{(1)}$ and $x^{(2)}$, : $D_0(x^{(1)}, x^{(2)}) - \text{MAX}_{c,c\neq0}(D_c(x^{(1)}, x^{(2)})) \geq \mathcal{E}^{raw}_{c^{(1)}}(x^{(1)}) - \text{MAX}_{c,c\neq c^{(1)}}\mathcal{E}^{raw}_c(x^{(1)}) + \mathcal{E}^{raw}_{c^{(2)}}(x^{(2)}) - \text{MAX}_{c,c\neq c^{(2)}}\mathcal{E}^{raw}_c(x^{(2)})$.

As $\text{MAX}(a,b) \leq \text{MAX}(a) + \text{MAX}(b)$, $\text{MAX}_{c,c\neq0}(D_c(x^{(1)}, x^{(2)})) \leq \text{MAX}_{c,c\neq c^{(1)}}\mathcal{E}^{raw}_c(x^{(1)}) + \text{MAX}_{c,c\neq c^{(2)}}\mathcal{E}^{raw}_c(x^{(2)})$. Thus negating and adding $D_0(x^{(1)}, x^{(2)})$ to both sides, we arrive at our claim. Hence our claim is true.

As our claim is true for all $x^{(1)}$ and $x^{(2)}$ and $\forall_x f(x) \geq g(x) \implies \text{MAX}_x(f(x)) \geq \text{MAX}_x(g(x))$, our objective is an upper bound over the ideal objective.

Thus, we can safely say if VERITAS outputs a value less than $2 * \eta$ or it timeouts, while our tool gives a sat output, our tool is better than VERITAS . If our tool gives sat but VERITAS provides a higher output, we deem VERITAS to be better. If our tool gives unsat, then we ignore that instance.

We give VERITAS 1200 seconds to run for each experiment and compare with the best output found till then. For all other tools, we compare time taken for them to find a satisfying pair of examples. The results of the experiments are given in Table 2. The Veritas algorithm finds progressively larger and larger gaps. %V indicates the amount of "gap" found by Veritas during the given time as compared to ENSENSE. For instance, consider the Iris row where we report 2%, which implies that the gap found by ENSENSEis 50 times bigger than the gap found by Veritas.

## G    ADDITIONAL EXPERIMENTS AND DETAILS

In this section, we provide more details regarding how we trained the models, how we performed our experiments, and also present additional experimental results. We then explain the counterexample region that is used to evaluate the distances from data. Then we do an ablation study to understand the impact of each improvement both in binary and multiclass tree ensembles. Finally, we show what happens when the sensitive feature set is larger than 1, say 2-4.

| Dataset | #Class | Dep. | #Trees | ENSENSE | KANT | %V |
|---|---|---|---|---|---|---|
| covtype_robust | 10 | 6 | 100 | 139.08 | 3086.10 | 0 |
| covtype_unrobust | 10 | 6 | 100 | 213.78 | 3087.16 | 0 |
| fashion_robust | 10 | 6 | 100 | 118.76 | 5667.60 | 0 |
| fashion_unrobust | 10 | 6 | 100 | 67.63 | 5001.60 | 0 |
| ori_mnist_robust | 10 | 6 | 100 | 108.76 | 3343.97 | 0 |
| ori_mnist_unrobust | 10 | 6 | 100 | 76 | 3587.97 | 0 |
| Iris | 3 | 1 | 100 | 0.01 | 0.01 | 2 |
| Red-Wine | 3 | 6 | 100 | 3.83 | 3.89 | 100 |

Table 2: Multi-Class comparison experiments with VERITAS and KANT. The table reports PAR2 runtimes for the experiments in Fig. 3, counting any timeout as $2 \times$ the timeout.

### G.1 TRAINING DETAILS

We trained XGBoost (v1.7.1; binary:logistic) with label encoding for categoricals, rows with missing values removed, and hyperparameters chosen on a 20% validation split (seed = 42) over maximum depth$\in \{5,6\}$ and number of boosting rounds $\in \{200,300,500\}$ (benchmarks 7–9) or 500,800 (benchmark 10).

### G.2 ABLATION STUDY

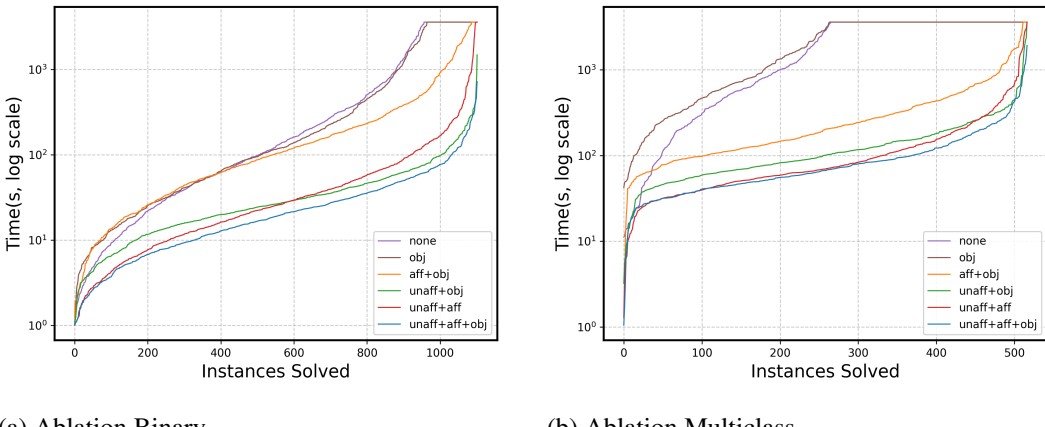

(a) Ablation Binary        (b) Ablation Multiclass

Figure 6: Timing performance of single feature sensitivity for (a) binary ensembles , and (b) multiclass ensembles.

To evaluate the contribution of each component in our sensitivity analysis, we conducted an ablation study by systematically removing key optimizations equation UnAff and equation Aff-bin and evaluating the resulting performance. We present these results in Figure 6, for binary tree ensembles in (left) and multi-class tree ensembles (right). Figure6(left) reports the results for 1102 benchmarks whose runtime is >1s and omitting 188 instances solved under 1s. Figure6(right) reports the results for 517 benchmarks whose runtime is >1s, omitting 21 instances solved under 1s Overall, the added constraints improve solver performance by up to an order of magnitude and dramatically reduce the number of timed-out instances. An interesting observation in both these plots is that when equation Aff-bin is added then equation UnAff do not contribute much in the performance (as can be seen by the overlapping lines). Overall, these results confirm that our enhancements significantly improve the practical feasibility of sensitivity verification in binary and multiclass decision tree ensembles.

### G.3 MULTIFEATURE SENSITIVITY ANALYSIS

To evaluate the ENSENSE's ability to handle multi-feature sensitivity (i.e., sensitivity wrt change in more than one feature simultaneously $|F| > 1$), we conducted experiments on binary classifica-

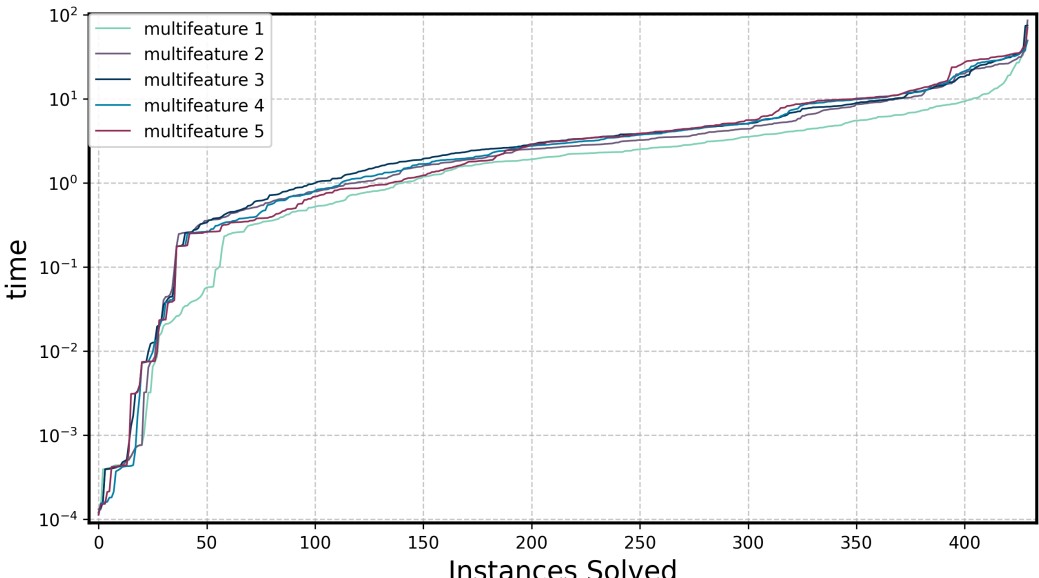

Figure 7: Timing performance of Multifeature sensitivity for Binary Classification

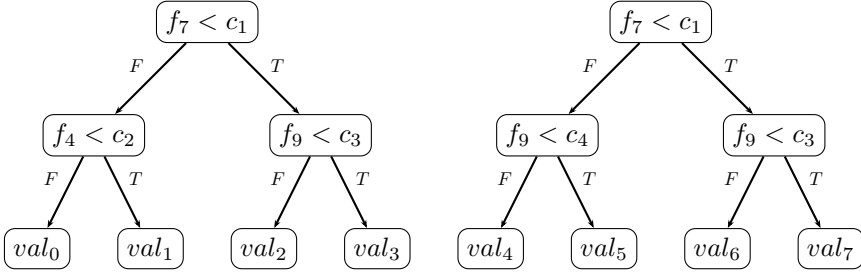

Figure 8: A tree ensemble with two trees $T_1, T_2$ having real valued (raw) outputs on leaves

tion models, allowing 1, 2, 3, 4, and 5 features to vary simultaneously. For each benchmark and each $m$-feature(s) setting, we generate as many test instances as the total number of features. Each instance corresponds to a randomly sampled subset of $m$-feature(s) from the feature set. Across all benchmarks (binary classifier) in Table 1, this results in a total of 430 instances for the $m$-feature(s). The results, shown in Figure 7, demonstrate that even as the number of varying features increases, our tool remains scalable and even improves in performance. The reason is that the search space explored by the tool decreases as we increase number of sensitive features (since we search in the space of $\mathcal{F} \setminus F$). These results demonstrate the framework's scalability and effectiveness in performing multi-feature sensitivity analysis.

## H    AN ILLUSTRATIVE EXAMPLE

In Figure 8, we show a tree ensemble with two trees $T_1$ and $T_2$. Each tree has four leaves with real-valued outputs $val_i, i = 0, \ldots, 7$. Let us assume that the sensitive feature set is $\{f_4\}$. We want to verify if there exists two inputs $x^{(1)}$ and $x^{(2)}$ differing only in feature $f_4$ such that the output of the ensemble changes from $0.5 - gap$ to $0.5 + gap$.

To formulate this as a MILP, we introduce the following variables:

- $p_{41}^{(1)}, p_{71}^{(1)}, p_{91}^{(1)}, p_{92}^{(1)}$: binary variables to represent the decisions at the internal nodes of the trees for input $x^{(1)}$.

- $p_{41}^{(2)}, p_{71}^{(2)}, p_{91}^{(2)}, p_{92}^{(2)}$: binary variables to represent the decisions at the internal nodes of the trees for input $x^{(2)}$.
- $l_0^{(1)}, l_1^{(1)}, l_2^{(1)}, l_3^{(1)}$: binary variables indicating which leaf of tree $T_1$ is reached by input $x^{(1)}$.
- $l_4^{(1)}, l_5^{(1)}, l_6^{(1)}, l_7^{(1)}$: binary variables indicating which leaf of tree $T_2$ is reached by input $x^{(1)}$.
- $l_0^{(2)}, l_1^{(2)}, l_2^{(2)}, l_3^{(2)}$: binary variables indicating which leaf of tree $T_1$ is reached by input $x^{(2)}$.
- $l_4^{(2)}, l_5^{(2)}, l_6^{(2)}, l_7^{(2)}$: binary variables indicating which leaf of tree $T_2$ is reached by input $x^{(2)}$.

The objective function in the following is equation Obj-bin for our example.

$$
\begin{aligned}
\max\ & val_0\, l_0^{(1)} + val_1\, l_1^{(1)} + val_2\, l_2^{(1)} + val_3\, l_3^{(1)} + val_4\, l_4^{(1)} + val_5\, l_5^{(1)} \\
& + val_6\, l_6^{(1)} + val_7\, l_7^{(1)} - val_0\, l_0^{(2)} - val_1\, l_1^{(2)} - val_2\, l_2^{(2)} \\
& - val_3\, l_3^{(2)} - val_4\, l_4^{(2)} - val_5\, l_5^{(2)} - val_6\, l_6^{(2)} - val_7\, l_7^{(2)}
\end{aligned}
\tag{14}
$$

The above objective is subject to the following constraints. In the guards of $f_9$, we assume $c_4 < c_3$. Therefore, the following constraints are due to Equation 2.

$$
\begin{aligned}
p_{91}^{(1)} &\leq p_{92}^{(1)} \\
p_{91}^{(2)} &\leq p_{92}^{(2)}
\end{aligned}
\tag{15}
$$

The following constraints are due to Equation 3.

$$
\begin{aligned}
l_0^{(1)} + l_1^{(1)} + l_2^{(1)} + l_3^{(1)} &= 1 \\
l_0^{(2)} + l_1^{(2)} + l_2^{(2)} + l_3^{(2)} &= 1 \\
l_4^{(1)} + l_5^{(1)} + l_6^{(1)} + l_7^{(1)} &= 1 \\
l_4^{(2)} + l_5^{(2)} + l_6^{(2)} + l_7^{(2)} &= 1
\end{aligned}
\tag{16}
$$

The following constraints are due to Equations 4 and 5.

$$
\begin{aligned}
-p_{71}^{(1)} + l_0^{(1)} + l_1^{(1)} &= 0 & \qquad -p_{71}^{(2)} + l_0^{(2)} + l_1^{(2)} &= 0 \\
p_{71}^{(1)} + l_2^{(1)} + l_3^{(1)} &= 1 & p_{71}^{(2)} + l_2^{(2)} + l_3^{(2)} &= 1 \\
-p_{71}^{(1)} + l_4^{(1)} + l_5^{(1)} &= 0 & -p_{71}^{(2)} + l_4^{(2)} + l_5^{(2)} &= 0 \\
p_{71}^{(1)} + l_6^{(1)} + l_7^{(1)} &= 1 & p_{71}^{(2)} + l_6^{(2)} + l_7^{(2)} &= 1 \\
-p_{41}^{(1)} + l_0^{(1)} &\leq 0 & -p_{41}^{(2)} + l_0^{(2)} &\leq 0 \\
p_{41}^{(1)} + l_1^{(1)} &\leq 1 & p_{41}^{(2)} + l_1^{(2)} &\leq 1 \\
-p_{92}^{(1)} + l_2^{(1)} &\leq 0 & -p_{92}^{(2)} + l_2^{(2)} &\leq 0 \\
p_{92}^{(1)} + l_3^{(1)} &\leq 1 & p_{92}^{(2)} + l_3^{(2)} &\leq 1 \\
-p_{92}^{(1)} + l_6^{(1)} &\leq 0 & -p_{92}^{(2)} + l_6^{(2)} &\leq 0 \\
p_{92}^{(1)} + l_7^{(1)} &\leq 1 & p_{92}^{(2)} + l_7^{(2)} &\leq 1 \\
-p_{91}^{(1)} + l_4^{(1)} &\leq 0 & -p_{91}^{(2)} + l_4^{(2)} &\leq 0 \\
p_{91}^{(1)} + l_5^{(1)} &\leq 1 & p_{91}^{(2)} + l_5^{(2)} &\leq 1
\end{aligned}
\tag{17}
$$

The following constraints are due to Equation 6.

$$
\begin{aligned}
-p_{91}^{(1)} + p_{91}^{(2)} &= 0 \\
-p_{92}^{(1)} + p_{92}^{(2)} &= 0 \\
-p_{71}^{(1)} + p_{71}^{(2)} &= 0
\end{aligned}
\tag{18}
$$

The following constraints are due to equation Gap-bin:

$$val_0 \, l_0^{(1)} + val_1 \, l_1^{(1)} + val_2 \, l_2^{(1)} + val_3 \, l_3^{(1)} + val_4 \, l_4^{(1)} + val_5 \, l_5^{(1)} + val_6 \, l_6^{(1)} + val_7 \, l_7^{(1)} \geq gap - 0.5$$
$$val_0 \, l_0^{(2)} + val_1 \, l_1^{(2)} + val_2 \, l_2^{(2)} + val_3 \, l_3^{(2)} + val_4 \, l_4^{(2)} + val_5 \, l_5^{(2)} + val_6 \, l_6^{(2)} + val_6 \, l_7^{(2)} \leq -0.5 - gap$$

$$(19)$$

The following constraints are due to Equations UnAff and Aff-bin respectively, since only feature $f_4$ is sensitive.

$$l_2^{(1)} = l_2^{(2)}$$
$$l_3^{(1)} = l_3^{(2)}$$
$$l_4^{(1)} = l_4^{(2)}$$
$$l_5^{(1)} = l_5^{(2)}$$
$$l_6^{(1)} = l_6^{(2)}$$
$$l_7^{(1)} = l_7^{(2)}$$

$$(20)$$

$$val_0 \, l_0^{(1)} + val_1 \, l_1^{(1)} - val_0 \, l_0^{(2)} - val_1 \, l_1^{(2)} \geq 2 * gap \qquad (21)$$

The above constraints indicates the key reason of the efficiency of our method. The long sum of Equation 19 reduces to much shorter sum . Thereby, MILP solver has easier time solving the problem.

