# OpenReview forum: "Data-Aware and Scalable Sensitivity Analysis for Decision Tree Ensembles"
_ICLR.cc/2026/Conference — ICLR 2026 Poster_

### Official Review · Reviewer_XHrk · 2025-10-27

**Soundness:** 2
**Presentation:** 1
**Contribution:** 3
**Rating:** 4
**Confidence:** 3

**Summary:**

This paper proposes a new method for the sensitivity analysis of decision tree ensembles.
Given a tree ensemble classifier, it aims to identify two data points that are close but receive different predictions from the model.
The authors also introduce an additional constraint that encourages two data points to be close to the dataset and propose an efficient algorithm based on mixed-integer linear programming (MILP).
By experiments with large tree ensemble models, the authors demonstrated that the proposed method achieved significant speedups over baseline methods.

**Strengths:**

1. This paper tackles an important problem of sensitivity analysis for tree ensemble models. The authors provide a theoretical analysis for the case that has not been addressed in prior work (Ahmad et al., 2025).
1. The experimental results demonstrate the scalability of the proposed method. Figure 3 clearly demonstrates that the proposed method could obtain solutions in a reasonable time compared to the baselines.

**Weaknesses:**

1. The presentation of Section 5 could be improved entirely. It is difficult to distinguish which terms are decision variables and which are fixed parameters in the proposed MILP formulation, making it hard to implement and reproduce. Adding clear notation definitions and a small illustrative example would help readers understand the formulation more easily and grasp the intuition behind the constraints.
1. While Figure 4 appears to show the L2 distance of each method for each instance, it is difficult to interpret and extract meaningful insights from it. It might be more precise and informative to summarize the results in a table reporting average values or win rates, rather than presenting all instance-level values in a single figure.
1. (Minor) The motivation of “data-awareness” has also been discussed in the literature on counterfactual explanations, which aim to generate realistic perturbations to alter prediction outcomes (e.g., Cui et al., 2015; Kanamori et al., 2020; Parmentier and Vidal, 2021). It would be helpful if the authors could clarify how their notion of data-awareness differs from or extends these prior concepts.

**Questions:**

1. The proposed method aims to generate pairs of data points $x^{(0)}$ and $x^{(1)}$ that are close to the dataset $\mathcal{D}$. Why not simply use data points already contained in $\mathcal{D}$? If one of the points, say $x^{(0)}$, were fixed to a sample from $\mathcal{D}$, can we formulate the problem as a more tractable one while still achieving the same objective?
1. Could the authors clarify which part of the analysis in the prior work (Ahmad et al., 2025) makes it impossible to extend or adapt to the cases with depth 1 or 2? A brief explanation would help contextualize the limitation and highlight the novelty of the proposed approach.
1. In the experiments, both the proposed method and Kant rely on an MILP solver to obtain solutions. Which solver did the authors use?

---

> ### Author Response · Authors · 2025-11-20
>
> Question 1
>
> We believe these are different questions. Our basic premise is that we are given a decision tree ensemble and a data set (potentially the training data set but not necessarily so), and our problem (lets call it A) is to ask if there is a pair of points that are close to data but have different answers. If we insist that one of the points actually must belong to the data set (lets call this problem B), then problem B is simpler (we just iteratively do a search over the dataset and for each point search in its neighbourhood), but this does not answer problem A. In particular, it may well be that the answer to B is no, but A is yes. From a problem modeling perspective, it is being close to data that makes the problem interesting. We believe that asking it to match the data, even for one point precisely, makes it a completely different objective and problem.
>
> Question 2
>
> Thanks for this question. In prior work (Ahmad et al., 2025), the authors perform a reduction from 3-CNFSAT to show NP-hardness by encoding each 3-CNF clause as a single depth-3 tree, with one literal tested on each level. So this reduction critically needs depth 3 trees. Note that 2-CNFSAT (which naturally come from adapting their reduction to depth 2) is solvable in polytime so useless for showing NP-hardness. Indeed, (Ahmad et al., 2025)  explicitly remark in Section 3 (last para of pg 6), that their construction relies on depth-3 trees and cannot be extended to depth-1 (or depth-2). Our proof in this work sidesteps this completely by providing a new reduction, from a different problem, namely the subset sum problem which is also NP-hard but  which now  allows us to obtain hardness for ensembles of decision stumps (i.e., decision 1 trees).
>
> Question 3
>
> We use the Gurobi solver ( Gurobi Optimization, LLC (2024) ). We have added this in the revised version (in the Experiment section)
>
> Regarding adding a "Worked out example and notation table."
>
> We thank the reviewer for this very nice suggestion. We have added a notation table in the appendix (see Appendix A) with a glossary of all terms in the revised version. Further, as advised, we have given a fully worked out example of our base encoding and affected and unaffected constraints, and the basic objective function in Appendix I. We have also improved the presentation of Section 5 in a few parts by improving notations and their explanations. We hope this makes things clear.
>
> Regarding "Improved Figure 4 and the presentation of results."
>
> Thanks! We have added average value details and win rates pairwise comparison across all methods in Figure 4. We hope this makes it easier to interpret our results and the effectiveness of our overall approach and the various parts thereof. We also note that during re-analysis, we fixed a minor plotting error (a few points had been erroneously added due to a plotting bug, which we removed). We have added all the deaggregated data in the supplementary material.
>
> Regarding "Discussion on the related literature."
>
> Thanks for these references. Indeed, the main difference is that, like for local robustness, these counterfactual explanations are given with respect to a given input point, while our notion of sensitivity is quantified over the entire input space of the model – in that sense, it is global and hence a more difficult search problem (as we have written in our comparison with local robustness). Having said that, we believe that some of the notions with respect to data-awareness, e.g., the Mahalanobis distance, could be useful for our problem as well as they have already used an MILP encoding for this. Also, we would like to point out that our approach of finding cavities uses SAT/SMT solvers instead of MILP and hence is orthogonal and likely finds/eliminates different cavities from the outliers that these earlier approaches detected. We will add a section on related work regarding this discussion.

---

> > ### Comment · Reviewer_XHrk · 2025-11-26
> >
> > Thank you for taking the time to answer all my questions and address all my concerns.
> > In particular, I would like to thank you for adding the notation table, working example and winning rate table to the revised version.
> > As some of my concerns were addressed in the rebuttal, I have decided to increase my score.
> > If possible, I hope the discussion on questions 1 and 2 will be included in the final version.

---

> > > ### Author Response · Authors · 2025-11-27
> > >
> > > Thank you again for positive feedback and also for the many constructive suggestions, which helped improve the paper. We will certainly add the discussions on questions 1 and 2 in the final version of the paper. If there are any  other concerns that you feel were not sufficiently addressed or could further increase your support for the paper, please feel free to let us know. Thanks!

---

### Official Review · Reviewer_6zFs · 2025-10-28

**Soundness:** 3
**Presentation:** 3
**Contribution:** 3
**Rating:** 4
**Confidence:** 4

**Summary:**

This paper introduces several improvements, both in runtime and example quality, to the framework of Kantchelian et al. (2016) for finding pairs of sensitive points (or two inputs which are close to the dataset, and have very different probability predictions from a tree ensemble). The closeness to the dataset is one improvement, for which two methods are discussed. The first is an approach that measures a point's closeness to the dataset by the product of its marginals, and the second restricts the search to remove "cavities", or boxes in input space with no points. The authors also present several runtime optimizations based on adding additional constraints of consistency across predicate variables and leaf conditions. There are experimental results demonstrating improved performance on several binary and multiclass datasets, and a variety of parameter configurations.

**Strengths:**

1. The papers organization is clear and logical, and the paper is well situated in existing literature. It is clear from the authors' presentation what is novel, how it is novel, and what research problems each component of the paper are trying to address.

2. The new constraints that speed up optimization are well-reasoned and proven to be correct.

3. The ablation study is convincing that the combination of all methods leads to a greater improvement than any subset.

4. I am not an expert in this area, but I appreciate the effort to go above and beyond to compare to VERITAS by modifying this system to suit the sensitivity problem.

5. The experiments seem to demonstrate a runtime improvement over existing methods, as well as finding examples closer to the training distribution.

**Weaknesses:**

1. The benchmarking results are too aggregated and are lacking statistical significance testing or error bars. The results should have been obtained over repeat trials for each configuration, with a corresponding mean and standard deviation. Moreover, it seems that most problem instances evaluated in Figure 3 are on only 4 datasets -- the ones with a lot of combinations of # trees and depths. I think that this work requires 1) statistical significance testing on the results over repeat trials for all results, and 2) disaggregated versions of the results by dataset (perhaps in the appendix; the aggregated results are fine as long as the disaggregated versions are accessible).

2. Figure 4 is really hard to read, and the definition of distance to the dataset is unclear. I assume that the distance to the dataset is defined by the distance to the nearest training point. I am concerned that, under this definition, a counterexample could be close to only a single point far off in a low-density region of space but still appear "close" to the dataset.

3. The use of cavities to restrict the search space is not compelling to me. Even in the provided example, it is odd that we ignore potential counterexample pairs close to the boundary of the green box. Those points are just as "in distribution" as any other. Furthermore, at least by inspection, it seems to me that we could form overlapping cavities over the entire search space, and thereby restrict our search only to exactly the points in the training dataset (for which there can obviously be no counterexample pairs).

4. The examples of counterexample pairs in the appendix are not informative -- there is no relative scale to understand what different distance magnitudes mean, and there are no feature names to understand the actual relevance of the features involved. The conclusions in the paragraphs in section B in general ("quite far from any possible realistic data point and may not be very helpful", for example) are not well-justified and contextualized in terms of the actual features in the data.

5. The experiment in Figures 6 and 7, which do the ablation and vary the number of features to be searched over, seem to be conducted on different datasets (fewer instances) than the other experiments. These should be described.

**Questions:**

1. Are any of the approaches you're comparing against designed for multi-threaded systems? Is your approach suited for multi-threaded systems? I am skeptical of a single-core evaluation of MILP-solver based systems, which to my understanding (not my field, so please pardon the question) are designed to be parallelizable.

2. How do you place a limit on cavity constraint creation so that you don't cover the entire input space?

3. What is the relationship between this topic and the domain of adversarial example search, where the goal is to find an adversarial example very close (within some small $\varepsilon$) to a specific point in the dataset?

4. Does your approach model binary/categorical and continuous features in the same input space? Does this affect your distance-based methods? Modeling these in the same input space enforces an assumption about the relative "cost" of flipping a binary variable and moving around on a continuous feature.

5. Very minor point, but the table headers in table 2 in the appendix are confusing. What is %V?

---

> ### Author Response · Authors · 2025-11-21
> **Addressing Direct Questions by the Reviewer: Comparing single and multi-threaded, cavity constraints, adverarial search and handling categorical features .**
>
> Question 1
>
> In Figure 3(a), we report the results on a single core, since the state of the art competing tool SensPB was a single threaded tool. However, to answer your question regarding suitability and use of multi-threaded systems, we performed a new set of experiments whose results are in Figure 9 of the Appendix. In particular, we ran our tool afresh in a more powerful 256 core machine in single and multithreaded modes (Figure 9) and compared the results. The results and observations presented in Appendix H are interesting. For benchmarks which are solved fast (e.g., less than 10sec), there is little or no difference. However for benchmarks that take longer time (>10s) the multi-threaded curves start dominating the single-threaded ones.
>
> Question 2
>
> Thanks for raising this important point. Our cavity constraints can’t cover the entire input space because (i) constraints are not learned on the data points but over the splits present in trees only, ii) each cavity is restricted to a clause width ‘w’ (for our experiments w=3), limiting how many features it can involve, and (iii) we impose the limit (for our experiments, we set 1500) on the number of learned cavities. Through all these we ensure that we don’t end up covering the entire input space.
>
> Question 3
>
> Adversarial-example search corresponds to local robustness, where the goal is to find an epsilon-perturbation around a fixed input that flips the prediction. A large body of work studies this setting for decision trees (cited in Related Section) where it is often formulated as local robustness. Our problem differs fundamentally: sensitivity verification requires reasoning over all pairs of inputs (universal quantification over two points), not just small neighborhoods around a given sample, making it more complex. We have also included a discussion on this distinction in the Related Work Section of the paper.
>
>
> Question 4
> Thanks for this question. Yes, we have considered this issue and it does play a role. Ideally, we believe that a domain expert should assign an appropriate “cost” for flipping each binary feature and that this cost should be used in the distance metric. As we ran our experiments without the availability of the domain expert, we chose to normalize all features, including categorical ones, to the range [0,1],  even if this causes categorical features to contribute disproportionately to the distance metric. For the categorical features, we applied the default label encoding available in the dataset. In the revised version, we have also explained this in more detail in the case of a particular benchmark(in Example 4 in Appendix C).
>
> Question 5
>
> The Veritas algorithm finds progressively larger and larger gaps. %V indicates the amount of “gap” found by Veritas during the given time as compared to our tool. For instance, consider the row where we report 2%, which implies that the gap found by SViM is 50 times bigger than the gap found by Veritas. We have added this explanation in the paper.

---

> > ### Comment · Reviewer_6zFs · 2025-11-21
> >
> > Question 1: This is interesting and valuable to know, thank you for running this experiment. My primary concern was that competing methods were potentially hamstrung, and that concern has been assuaged.
> >
> > Question 2: Am I right, then, to understand cavities as regions of the partition of the input space induced by the ensemble (up to 3 splits) which contain no points? How are the 1500 cavities determined (i.e. what order do you build the list of cavities)?
> >
> > The follow up questions will help me to understand your method better -- as far as I can observe, these constraints are not described in the paper. I would recommend this discussion to be added to the appendix, perhaps with some justification as to the choices of a width of 3 and number of cavities 1500.
> >
> > Question 3: This makes sense, thank you.
> >
> > Question 4: Thank you for the discussion, and your response makes sense. I think that this issue is relevant enough to warrant brief discussion in a more visible section of the paper, but I will leave this to the authors' discretion.
> >
> > Question 5: Thank you.

---

> ### Author Response · Authors · 2025-11-21
> **Addressing Weaknesses: Disaggregated results added to Supplementary Material, Clarified experiments and added tables showing effective improvement.**
>
> In addition to our earlier response, we would like to address some of the remaining issues pointed out.
>
> Regarding “Statistical significance and aggregated results.”
>
> Please note that our results are not statistical but deterministic. As a result when we run the solver twice on same machine, we obtain *almost exactly* the same result (upto hardware noise); note that we also removed <1s runtime results from the plots. This is why we did not provide statistical significance testing. But we have indeed collected the disaggregated results by dataset and are happy to provide them. Since they were too large to be added the the appendix, we have added these results to the supplementary material. We hope they are accessible. In the final version, we will run our experiments three times and provide all the runtimes (and also provide our code in a public repository).
>
> Regarding “Congested figure 4…”
>
> In Figure 4, the distance from the data to a counterexample region is defined as the minimum Euclidean distance between any training point and counterexample region, computed only over the non-sensitive features (i.e., we ignore the sensitive feature when evaluating this L2 distance). But the concern raised is valid—nearest-neighbor distance can regard a point as close even when it lies near only a single isolated sample in a low-density region. Nevertheless, the presence of that training point still provides evidence that the discovered counterexample arises near a region supported by real data.
>
> Regarding “Data Points close to the boundary”
> There are two points raised by the reviewer here. On one hand, the reviewer is right that points near the boundary of a cavity may still be “in-distribution.” Our use of cavities is a heuristic intended only to prune regions that do not cover the data. To avoid the case described, we can allow ingress of some epsilon in the cavity, and we consider adding such softened boundaries as an interesting direction for future work.
> On the other hand, as also mentioned in the answer to reviewer “6zFs”, our cavity constraints can overlap but they can’t cover the entire input space because of three reasons: (i) constraints are not learned on the data points but over the splits present in trees only, ii) each cavity is restricted to a clause width ‘w’ (for our experiments w=3), limiting how many features it can involve, and (iii) we impose the limit (for our experiments, we set 1500) on the number of learned cavities.
>
> We highlight that our heuristical approach is indeed effective in practice, as shown by our experimental results. Beyond the previously presented experiments, we have now added *additional experiment details* in Figure 4 in Section 7, where we report the mean distance from data for each method. Here ‘probclause’ (our approach that involves both utility function and cavity search) is clearly the best, and reports the minimum among all the compared methods, with mean distance of just 0.17 to the data. In addition, in Figure 4, we have added a comparison between every two of these methods with each other and report the % of benchmarks where the first wins (i.e., gives a closer counterexample to data) or loses (with a draw meaning that counterexample has the same distance). In short, the results show that using the ‘clause’ method only already results in a 80.5% win over ‘none’ and our best ‘probclause’ method wins with 86.74% against ‘none’ and loses only with 12.1%.
> We hope with these tables and explanations, it is clear that both our approaches help and are complementary to each other, such that adding them together gives the best benefits.
>
> Regarding “Contextualized counterexample”
>
> We have added the feature names with their original values and the nearest datapoint for examples 3, 4, and 5. For examples 1 and 2, the original feature names are unfortunately unavailable, as these benchmarks were taken from “Robustness Verification of Tree-Based Models” (Chen et al., 2019), and the corresponding dataset with named features is not provided along with the data.
>
> Regarding “Mismatch in the number of benchmarks”
>
> Thank you for pointing this out! In Figures 6 and 7, we apply the same filtering as in Figure 3 which we had mentioned : instances completing in under 1 second are omitted. This excludes 188 trivial cases, leaving 1,102 benchmarks in Fig. 6(a), and excludes 21 cases, leaving 517 benchmarks in Fig. 6(b). We have addressed this in the revised version under section “G.3 Ablation Study”.
> For Fig. 7 (the multi-feature setting), we have added additional experiments for multifeature sensitivity analysis in Figure 8. For each benchmark and each m-feature(s) setting, we generate as many test instances as the total number of features. Each instance corresponds to a randomly sampled subset of m-feature(s) from the feature set. Across all benchmarks (binary classifier) in Table1, this results in a total of 430 instances for the m-feature(s).

---

> > ### Comment · Reviewer_6zFs · 2025-11-21
> >
> > ## Regarding “Statistical significance and aggregated results.”
> >
> > This is satisfactory, thank you for including the disaggregated results in the supplementary material. And I think that including 3 runs and the corresponding runtimes will be useful to be confident in the conclusions, even (and especially) if the runtimes are all very similar to each other.
> >
> > ## Regarding “Congested figure 4…”
> >
> > I think that this distance measure should be described in more detail in the paper. It is relevant to understanding the conclusions of the experimental sections.
> >
> > The additional experimental discussion and the win-rate table are both very useful for building confidence in the empirical performance of your method. Thank you for including these.
> >
> > ## Regarding “Contextualized counterexample”
> >
> > Thank you.
> >
> > ## Regarding “Mismatch in the number of benchmarks”
> >
> > Thank you for describing this.

---

> > > ### Comment · Reviewer_6zFs · 2025-11-21
> > >
> > > Following the extensive additional experiments added by the authors, and their additional description of their methods, I am much more confident in the experimental results and will happily raise my score to acceptance, given my already noted appreciation of the novelty and theoretical soundness of the provided algorithms for identifying counterexample pairs.

---

> > > > ### Author Response · Authors · 2025-11-22
> > > >
> > > > Thanks very much for your prompt response and for increasing the score and your support for the paper.
> > > >
> > > >
> > > > Regarding your followup question regarding cavities, indeed, cavities are exactly regions of the partitions of the input space induced by the ensembles which have no points, where width 3 in our experiments corresponds to 3 features, and hence we may get up to 6 splits (corresponding to upper and lower bound guard for each). The order in which the cavities are learnt is by order of the increasing width, and for each value of width, we use cavities in the order returned by the constraint solver (Z3 in our case).
> > > >
> > > >
> > > > Regarding the choice of the parameters (width 3, and no. of cavities 1500), our choice is driven by the tradeoff between solving time and effectiveness. If we add too many cavity constraints to the MILP solver, it will be overwhelmed by the constraints. So, we heuristically choose a number such that the performance improves without significantly degrading the speed of the solver. Note also that if we allow many features to appear in a cavity, we risk including very small cavities, and there may be many of these. To balance these objectives, we chose 1500 and 3, after trying out multiple combinations.
> > > > As suggested, we will add a discussion regarding these choices in the experimental section of the main paper itself. We will also try to add the discussion on handling categorical features for the distance measurement in the final version of the paper.
> > > >
> > > >
> > > > We thank you for these (and earlier) questions, which allowed us to clarify our method and experimental choices.

---

### Official Review · Reviewer_6CwP · 2025-10-30

**Soundness:** 4
**Presentation:** 3
**Contribution:** 3
**Rating:** 8
**Confidence:** 3

**Summary:**

This paper introduces efficient techniques to evaluate the sensitivity of an ensemble of trees to different subsets of features. The authors propose a specialized MILP formulation to construct counterexample pairs, while insuring that the constructed counterexample is realistic with respect to the data distribution. Empirical analysis shows that, across several datasets and models, this tool provides substantial runtime improvements over existing methods.

**Strengths:**

- The empirical results for SVIM are quite compelling. SVIM universally improves upon the runtime of prior methods, while producing counterexamples that are designed to be realistic.
- The proposed MILP formulation is interesting and involves several novel components.
- The problem studied here is interesting and well motivated.
- In general, the paper is well written and figures are clear.

**Weaknesses:**

- While the paper is generally easy to follow, it is quite notation heavy and suffers from some minor inconsistencies. In addition to the several specific comments listed below, I recommend adding a notation table to the appendix to help readers keep up.
    - In EQ 4/5, should $p_{kf}$ be $p_{fk}$?
    - In Eq Gap-Bin and Off-bin, $v_i$ is undefined.
    - In the data aware objective function defined under Utility Function, it seems like some things may be off. Should solution (2) be considered in some way, rather than just solution (1)? Additionally, I don’t think pi_f without an input value is well defined.
    - In the constraint introduced under Computing clause summaries, I believe w is undefined. Additionally, the subscript on the “and” is a bit off — should $j \in |F|$ be either $j \in F$ or $j \in [1, |F|]$?
- The L2 distance may not be an appropriate metric to measure counterexample validity because it may be thrown off if one entry in the original, unperturbed sample is extreme. Maybe an L_infininty distance would be more appropriate? This does depend on how features were pre-processed. Were they normalized in some way?

**Questions:**

- I would recommend adding the 1 feature case to Figure 7 to support a direct comparison.
- In the examples in the appendix, it would be nice to know what each of the reported features represents.
- I would consider this out of scope for the current work, but in the future I would recommend conducting a brief user study to see whether the counterexamples generated by this method are generally considered more realistic.

---

> ### Author Response · Authors · 2025-11-20
> **Clarified encoding and revised experiments**
>
> Question 1
>
> Thanks for the suggestion. We chose to keep Figure 7 as is, but we have added additional experiments with 1-feature case included; please see Figure 8 in the appendix of the revised version.
>
> Question 2
>
> This is a good idea; thanks! We have replaced the values (which indeed were difficult to understand) with the feature names in Examples 3,4, and  5 in Appendix C. For examples 1 and 2, the original feature names are unfortunately unavailable, as these benchmarks were taken from “Robustness Verification of Tree-Based Models” (Chen et al., 2019), and the corresponding dataset with named features is not publicly provided.
>
> Question 3
>
> Thanks for this excellent suggestion! We are, in fact, in the process of developing such a study. Some of this work is being deployed at a financial institution, where we apply our tool for analyzing their tree ensemble models and data sets. Surprisingly, in many cases, we observe an even better performance than what we present in this paper, presumably because of the specific characteristics of their dataset. However, due to privacy and corporate reasons, we cannot reveal or publish these results at this point. However, inspired by your comment, we will perhaps try to do a public user case-study (maybe among students in a university etc for a relevant example) in a way that can be published.
>
> Typographical errors
>
> We acknowledge and apologize for the typographical errors pointed out. We thank you for carefully reading our manuscript and identifying them. We have addressed them in the revised manuscript and also tried to simplify a few notations and avoid overloading. We have highlighted our changes in blue for your easy reference. We have also added a notation table in the appendix (see Appendix A) with a glossary of all terms, as advised. In particular, we would like to note that we had not provided the full encoding of the utility function in the paper (Section 5) in terms of the encoding variables and this led to some confusion. To fix this, we have now described the encoding of the function in Eq 7 of Section 5.3 and given a proof of why it captures our utility function in the appendix (Lemma E.1). Furthermore, in Appendix I we have given a fully worked out example of our base encoding and affected and unaffected constraints, and the basic objective function. We note that all this is consistent with what has been implemented in our code, which we fully intend to make public after acceptance to allow validation and repeatability.
>
> L2 vs L_infinity
>
>
> We thank the reviewer for raising this point. We agree with the reviewer that it would be good to experiment with different norms. However, in the presence of categorical features, we found that L_infinity was also susceptible to extreme changes. In our setting, we normalized all features to be between [0,1]. But in the future, we could also consider handling categorical and non-categorical features differently, and perhaps apply norms and weights that are appropriate and user-defined.

---

> > ### Comment · Reviewer_6CwP · 2025-11-24
> >
> > Thank you for the thorough response! I find the notation table and fixes to writing/notation helpful. I find it encouraging that such a user study is underway, and do encourage the authors to do some form of it that could be published at a later date.
> >
> > I recommend that the authors explicitly state somewhere in the appendix that features were normalized between 0 and 1 (apologies if this has already been done -- I looked, but didn't see it).
> >
> > I maintain that this is a strong, interesting, and sound work that should be accepted.

---

> > > ### Author Response · Authors · 2025-11-25
> > >
> > > Thank you for the positive feedback and helpful suggestions. We will explicitly add in Section H.1 of the appendix that all features are normalized to the range [0, 1]. We indeed appreciate your strong support towards acceptance of the paper.

---

### Official Review · Reviewer_KgVe · 2025-11-12

**Soundness:** 3
**Presentation:** 3
**Contribution:** 3
**Rating:** 6
**Confidence:** 2

**Summary:**

The paper studies the feature sensitivity problem in decision tree ensembles. The authors proposed a data-aware sensitivity framework that builds on the combination of MILP and SMT encoding to verify whether prediction changes under feature perturbations. They developed a method SVIM and demonstrated the effectiveness of their methods compared to prior work on empirical data sets.

**Strengths:**

1. The paper shows sensitivity verification is NP-hard even for depth-1 trees.
2. The authors implemented their method SVIM and outperformed prior baselines such as SENSPB and KANT.
3. They extended the method beyond binary classification to multi-class problems.
4. The paper is clear in definitions and theoretically well-justified.

**Weaknesses:**

1. Whether the “data-aware” counterexamples lead to practical improvements in model fairness or robustness is not well explored, and remains an interesting practical direction.
2. The independence assumption is mitigated through restricting space, yet it still introduces a theoretical gap between the assumed and real data distribution.

Minor formatting: “Figures” vs “Fig”. (Line 456)

**Questions:**

1. Could this method be generalized for regression problems?
2. Will the “data-aware” counterexample be used for model retraining, will there be model improvements observed?

---

> ### Author Response · Authors · 2025-11-20
> **Support for regression models and using counterexamples for model improvement**
>
> Question 1
>
> Thanks for this question. We believe the broad approach can be generalized to regression problems, where the output space is continuous. Most of the theory would lift immediately, as the core combinatorial problem that we address doesn’t change much. But in practice, we will need to work out the details, e.g., how to model the output gap. We believe this could be a very interesting future direction and a source for more benchmarks.
>
> Question 2
>
> While this work focuses on identifying counterexamples, the idea of data-awareness in this context is to identify more meaningful counterexamples, which can serve as feedback to model developers. Indeed, the logical next step would be to use these counterexamples for model retraining. While in theory we are sure (and we can prove this formally) that the retraining will improve the model, it remains to be seen how effective it will turn out to be in practice for large benchmarks.  This would depend on precisely how the retraining is done. For instance, if we introduce the sensitive points by assigning them values based on the average of their neighboring points, during training, it may not be practical, as identifying such points each time is computationally expensive. We therefore leave retraining as out of scope for this paper and in the realm of immediate but future work.
>
> Response to “The independence assumption is mitigated through restricting space, yet it still introduces a theoretical gap between the assumed and real data distribution.”
>
> Thanks for this comment. Indeed, our reason to introduce the cavity-based approach was to mitigate the independence assumption. This does not completely eliminate the theoretical gap, but we wish to highlight that our goal was twofold: obtain theoretical guarantees and have good practical performance. Indeed, it is likely that eliminating the theoretical gap precisely would lead to higher complexity in our solution (or not be compatible with current MILP solvers)  and hence not be scalable. Our approach tries to balance the fine line between theoretical guarantees and practical implementability as shown in our experiments. Indeed, our practical performance suggests that we are closer to the real data than our theoretical guarantees suggest in the benchmarks that we tested. However, we readily acknowledge that there may be settings where higher theoretical guarantees may be desirable at the expense of practical performance.

---

> > ### Comment · Reviewer_KgVe · 2025-11-25
> >
> > Thank you for your response!
> >
> > I agree that the generalization to regression problems and more practical aspects could be considered in future versions.

---

> > > ### Author Response · Authors · 2025-11-27
> > >
> > > Thank you so much for the positive feedback. We hope that you may also consider increasing the score in light of the feedback and response. Indeed, if there are more questions or suggestions that could potentially help in further strengthening your support, please feel free to let us know.

---

### Author Response · Authors · 2025-11-20
**Summary of our rebuttal**

We thank all the reviewers for their in-depth reading and useful suggestions and questions. In our response, we have addressed all questions and weaknesses pointed out. The main highlights of the response:
1. we have addressed the concerns related to the presentation of the paper, including fixing typos, expanding explanations and adding a glossary table of notations as suggested by a reviewer.
2. we have also performed some more experiments in response to the questions asked, especially for multi-feature and multi-threaded runs.
3. we have improved the presentation of experimental results (in Figure 4) via new tables,  on the lines of a suggestion by a reviewer.
4. we have also added supplementary material with disaggregated runs as requested by a reviewer.

The key contribution of our work is to develop an **efficient and data-aware** tool for the sensitivity problem, especially in light of our strengthened NP-hardness results. We have tried to highlight that our tool achieves this goal, via more explanations and additional experiments (described below) in response to the valuable feedback from the reviewers. All the changes in the paper are highlighted in blue. If there are any further questions, we are happy to answer.

---

### Author Response · Authors · 2025-11-30
**Post-rebuttal summary of our interaction with reviewers and resulting increases**

We were not aware of any hack until we received the email from the chairs, and are in extreme shock. We hope the new area chair will carefully evaluate our interactions with the reviewers and confirm that all reviewers now support our paper unanimously and that all the erstwhile increases in score were well and truly justified.

We would like to highlight the following points, which we believe also provide the clearest evidence for the absence of collusion at any point:

1. Our comment “Summary of our rebuttal” summarizes all the detailed changes we made, which addressed **every concern** of each reviewer. The revised version has clearly marked changes in blue, including new experiments conducted and newly uploaded supplementary material.

2. After our rebuttal response, ALL reviewers acknowledged our response and have explicitly expressed **unanimous and clear support**  for acceptance of our paper.

3. The two reviewers who had concerns earlier had both INCREASED their score, one from 4 to 8 on Nov 22nd and another from 4 to 6 on Nov 26th, both with clearly documented justifications. In fact, the reviewer who increased from 4 to 8 acknowledged our response to each of the negative concerns they had had and also mentioned that they “will happily raise my score to acceptance, given my already noted appreciation of the novelty and theoretical soundness of the provided algorithms…”.

4. The reviewer who already recommended acceptance, maintained in their post-rebuttal response that “this is a strong, interesting, and sound work that should be accepted”. We were in fact hoping that this reviewer may consider increasing their score to 10, given their strong support! Further in the normal course of affairs, we would have expected another reviewer to also potentially increase their presentation score given their post-rebuttal response expressing their satisfaction with our changes.

5. Finally, the reviewers suggested including the following specific points in the paper, which are all clearly marked in the rebuttal, and which we have promised to update accordingly.
- (6CwP and 6zfs) Add a discussion regarding handling of categorical features, choice of cavity learning parameters and normalization details.
- (Q1 and Q2 of XHrk) Add justification of our method of solving the problem and comparing with another approach suggested, and more details regarding the novelty of our NP hard proof.

Given all this (our revised scores were 8, 8, 6, 6 all with reasonable confidence and proper justifications, and still potential for further increase in scores), we were hoping not only for acceptance but hopefully an oral presentation. We are thus dismayed to find that we have to now restart from the earlier scores. We hope the new area chair will take into cognizance the effort that has already been put in and the consensus built, and judge the paper accordingly. Thanks for your patience and understanding.

---

### Meta-Review · Area_Chair_LKyv · 2026-01-08

**Summary:**

The paper addresses the problem of the sensitivity of tree ensembles to subsets of features. The Authors introduce an efficient approach based on mixed-integer linear programming (MILP) and satisfiability modulo theories (SMT) encodings to construct counterexample pairs that are close to examples in the training set. As a theoretical contribution, they strengthen the NP-hardness result for sensitivity verification by showing that it holds even for trees of depth 1. Empirical results demonstrate that the proposed approach achieves substantial runtime improvements over existing methods.

The Reviewers are generally positive about the contribution in their initial reviews. The main issues raised concern the practical usefulness of the “data-aware” counterexamples, a theoretical gap between the assumed and the true data distribution, limited experimental evaluation, clarity, and several typographical errors. The rebuttal adequately clarifies these concerns. Therefore, the paper should be considered for publication at the conference.

**Reviewer Concerns:**

The Authors adequately clarify the main concerns.

**Reviewer Scores:**

- KgVe: would likely keep the score of 6
- 6CwP: would likely keep the score of 8
- 6zFs: would likely increase the score to 5 or 6
- XHrk: would likely increase the score to 5 or 6

---

### Decision · Program_Chairs · 2026-01-26

Accept (Poster)